



# Advanced hodograph-based analysis technique to derive gravity waves parameters from Lidar observations

Irina Strelnikova[1], Gerd Baumgarten[1], and Franz-Josef Lübken[1]

[1]Leibniz-Institute of Atmospheric Physics at the Rostock University, Kühlungsborn, Germany

*Correspondence to:* Irina Strelnikova (strelnikova@iap-kborn.de)

**Abstract.** An advanced hodograph-based analysis technique to derive gravity waves (GW) parameters from observations of temperature and winds is developed and presented as a step-by-step recipe with justification of every step in such an analysis. As a most adequate background removal technique the 2D-FFT is suggested. For an unbiased analysis of fluctuation whose amplitude grows with height exponentially we propose to apply a scaling function of the form $\exp(z/(\varsigma H))$, where $H$ is scale height, $z$ is altitude, and the constant $\varsigma$ can be derived by a linear fit to fluctuation profiles and should be in a range 1–10 (we derived $\varsigma = 2.15$ for our data). The most essential part of the proposed analysis technique consist of fitting of cosines-waves to simultaneously measured profiles of zonal and meridional winds and temperature and subsequent hodograph analysis of these fitted waves. The novelty of our approach is that its robustness ultimately allows for automation of the hodograph analysis and resolves many more GWs than it can be inferred by manually applied hodograph technique. This technique is applied to unique lidar measurements of temperature and horizontal winds measured in an altitude range of 30 to 70 km. A case study of continuous lidar observations from January 09 to 12, 2016 with the ALOMAR Rayleigh-Mie-Raman (RMR) Lidar in Northern Norway (69°N) is analyzed. We use linear wave theory to identify 4507 quasi monochromatic waves and apply the hodograph method which allows to estimate several important parameters of the observed GW. This technique allows to unambiguously identify up- and downward propagating GW. In the vicinity of the polar night jet $\sim 30\ \%$ of the detected waves propagate downwards. The upward propagating GW predominantly propagate against the background wind, whereas downward propagating waves show no preferred direction. The kinetic energy density of upward propagating GW is larger than that of the downward propagating waves, whereas the potential energy is nearly the same for both directions. The mean vertical flux of horizontal momentum in the altitude range of 42 to 70 km for the detected waves is about 0.65 mPa for upward propagating GW and 0.53 mPa for downward propagating GW.

## 1 Introduction

It is generally accepted that atmospheric gravity waves (GW) produce global effects on the atmospheric circulation from the surface up to the mesosphere and lower thermosphere (MLT) region (e.g., Fritts and Alexander, 2003; Alexander et al., 2010; Becker, 2017). Well known tropospheric sources for these waves are the orography (flows over mountains), convection, and jet imbalance (e.g., Subba Reddy et al., 2005; Alexander et al., 2010; Mehta et al., 2017). When propagating upwards, GW dissipate and thereby deposit their momentum starting from the troposphere and all the way up to the MLT. This process is





referred to as GW-forcing and plays a key role in the global circulation. The problem is that most of the models are not able to resolve these small-scale waves. That is why these waves and their dissipation (and also their interaction with each-other and with the background flow) are often called "sub-grid scale processes" (e.g., Shaw and Shepherd, 2009; Lott and Millet, 2009). In order to account for the influence of GW they are forced to use various parameterizations. To construct a proper

parametrization one has to describe GW frequencies, wavelengths, and momentum flux over the model coverage zone (e.g., Alexander et al., 2010; Bölöni et al., 2016).

Our knowledge about gravity wave parameters can only be improved by means of high resolution measurements of atmospheric GW. Ideally, the measurement range should cover the entire path of the waves starting from their sources in the troposphere to the level of their dissipation, that is up to the MLT region. Such type of measurements ultimately faces high

experimental challenges which explains why we still do not have satisfactory and conclusive observational data on these processes.

In the altitude range of the mesosphere only few observation techniques exist. In the last decades the only source of high-resolution GW observations based on both temperatures and winds in the stratosphere and mesosphere region were rocket soundings (see e.g., Schmidlin, 1984; Eckermann and Vincent, 1989; Lübken, 1999; Rapp et al., 2002, and references therein).

Rocket measurements with e.g., falling spheres can provide vertical profiles of horizontal winds and atmospheric temperatures and densities with altitude resolution of about 1–10 km.

Satellite-borne remote sensing techniques can provide excellent global coverage, their observations deliver unique horizontal information about GWs (see e.g., Alexander et al., 2010; Alexander, 2015; Ern et al., 2018).

Ground-based radar systems are able to measure winds at heights 0–30 km and 60–100 km. From the altitudes between

30 and 60 km radars do not receive sufficient backscatter and, therefore cannot provide wind measurements in this region. While the vertical wave structure can be resolved from rocket profiles, the long and irregular time intervals between successive launches prevent the study of temporal gravity-wave fluctuations over a larger time span (Eckermann et al., 1995; Goldberg et al., 2004).

Recent developments in lidar technology give us new possibilities to study these small-scale waves experimentally on a more

or less regular basis (e.g. Chanin and Hauchecorne, 1981). In particular the day-light lidar capabilities allow for long duration wave observations (e.g., Baumgarten et al., 2015; Baumgarten et al., 2018). The new Doppler Rayleigh Iodine Spectrometer (DoRIS) additionally to the legacy lidar temperature measurements yields simultaneous, common volume measurements of winds (Baumgarten, 2010; Lübken et al., 2016). This combination of capabilities makes lidar data unique.

All those quantities, i.e. winds and temperature, when measured with high temporal and spatial resolution, reveal struc-

turing at scales down to minutes and hundreds of meters. These small-scale structures, hereafter referred to as fluctuations are produced by atmospheric gravity waves. By applying a proper data analysis technique one can extract several important parameters of GW from the advanced lidar measurements.

In this paper we describe a newly developed analysis technique which allows for derivation of GW parameters such as vertical wavelength, direction of propagation, phase speed, kinetic and potential energy and momentum flux from the advanced

lidar measurements. We aimed at presenting a step-by-step recipe with justification of every step in such an analysis. Every





single steps if considered independently, are in general well known. The strength and novelty of our work is their combination and some justification on their importance and how they affect analysis results. The paper is structured as follows. In the next section a short description of lidar measurement technique is given. Section 4 describes the new methodology in detail. Finally, in section 5 geophysically meaningful quantities are deduced from the analyzed data which also demonstrates the capabilities

of the introduced analysis technique. Theoretical basis used by the data analysis technique is shortly summarized in section 3 and extended in Appendix A.

## 2   Instrumentation

The ALOMAR Rayleigh-Mie-Raman lidar in northern Norway (69.3°N, 16.0°E) is a Doppler lidar that allows for simultaneous temperature and wind measurements in the altitude range of about 30 to 80 km. The lidar is based on two separate pulsed lasers

and two telescopes (von Zahn et al., 2000). Measurements are performed simultaneously in two different directions, typically 20 degrees off-zenith towards the North and the East by pointing the telescopes and the outgoing laser pulses in this direction. The diameter of each telescope is about 1.8 m and the average power of each laser is ∼14 W at the wavelength of 532 nm. Both pulsed lasers operate with a repetition rate of 30 Hz and are injection seeded by one single CW-laser that is locked to an Iodine absorption line. The light received by both telescopes is coupled alternatingly into one single polychromatic detection

system. Temperatures and winds are derived using the Doppler Rayleigh Iodine Spectrometer (Baumgarten, 2010). As the measurements discussed below are performed also under daytime conditions we process the data as described in Baumgarten et al. (2015). Measurements by the lidar were extensively compared to other instruments showing the good performance of the lidar system (Hildebrand et al., 2012; Lübken et al., 2016; Hildebrand et al., 2017; Rüfenacht et al., 2018). The lidar data are recorded with an integration time of 30 seconds and a range resolution of 50 m. The data are then integrated to a resolution of

5 minutes and 150 m and then smoothing with a Gaussian window with a full width at half maximum of 15 minutes and 0.5 km is performed. For calculation of horizontal winds from the measured line-of-sight winds we assume that the vertical wind component is equal to zero. Importantly, the estimated uncertainty imposed by this assumption is negligible and does not affect final results of our analysis. The hydrostatic temperature calculations were seeded using measurements from the IAP mobile Fe resonance lidar and the temperatures from both lidar systems were then combined by calculating an error weighted mean

(Lautenbach and Höffner, 2004).

## 3   Short theoretical introduction

A GW wave field consists of various waves with different characteristics. An attempt to describe this system as a whole is made, for example, by Stokes analysis (e.g., Vincent and Fritts, 1987; Eckermann, 1996)

In this work we do not try to describe bulk fluctuations, but rather to extract the single most dominant quasi monochromatic

(QM) gravity waves (GW) from the set of the observed fluctuations. The advantage of this approach is that it allows us to describe these selected waves as precisely as possible by the linear theory of GW. Moreover, the main idea of our retrieval is





to find GW-packets where fluctuations of the components $u'$, $v'$, and $T'$ show the same characteristics, i.e., belong to the same wave-packet. The latter requirement ensures, that our analysis only accounts for wave structures and not for those created by accompanying dynamical processes like turbulence or other wave-like structures created by e.g., temperature inversion layers (e.g., Szewczyk et al., 2013).

A Gravity wave may change its characteristics when propagating through a variable background, especially its vertical wavelength may change with height. In such a case our GW-retrieval technique detects several waves at different altitude ranges. At the same time, our analysis precisely describes the changing background so that it is clear which wave propagation conditions correspond to certain altitude ranges and, therefore, to the detected waves with certain parameters. In other words, even though our analysis does not capture all the existing waves, it allows us to investigate obtained wave characteristics as a

function of background properties.

For this analysis we use the assumption, that a wave packet at a fixed time point and in a limited altitude range can be considered as quasi monochromatic GW, i.e. dispersions within one wave packet is neglected. Also we assume, that all the observed parameters ($T'$, $u'$, $v'$) reveal fluctuations at the same wavelength. Mathematically it can be written in the following form (see Appendix A for more details):

$$T' = |\widehat{T}| \cdot \exp(-(z - z_0)^2/2\sigma^2) \cdot \cos(m(z - z_0) + \varphi_T) \cdot \exp(z/2H) \tag{1a}$$

$$u' = |\widehat{u}| \cdot \exp(-(z - z_0)^2/2\sigma^2) \cdot \cos(m(z - z_0) + \varphi_u) \cdot \exp(z/2H) \tag{1b}$$

$$v' = |\widehat{v}| \cdot \exp(-(z - z_0)^2/2\sigma^2) \cdot \cos(m(z - z_0) + \varphi_v) \cdot \exp(z/2H) \tag{1c}$$

where $T'$, $u'$ and $v'$ are fluctuations of temperature, zonal and meridional wind components and $\widehat{T}$, $\widehat{u}$ and $\widehat{v}$ are amplitudes of those fluctuations, $\sigma$ is a factor describing width of wave packet, $z_0$ is the altitude of maximum wave envelope, $\lambda_z = 2\pi/m$

is vertical wavelength, $\varphi_T$, $\varphi_u$, $\varphi_v$ are phase shifts of temperature, zonal and meridional wind fluctuations respectively, $m$ the vertical wave number and $H$ is the scale height.

This set of Eqs. 1 is an ansatz which describes a wave packet of an ideal monochromatic GW under the conditions of conservative propagation in a constant background. Similar description of GW propagation is widely used in the literature (see e.g., Gavrilov et al., 1996), which we extend by introducing a wave packet envelop term $\exp(-(z - z_0)^2/2\sigma^2)$ that accounts

for limited presence of the GW-packet in observations.

In the center of the wave packet we apply the hodograph analysis to extract the essential parameters of the wave packet (e.g. Baumgarten et al., 2015). Fig. 1 schematically illustrates this method. In the center of the wave packet the QM wave produces fluctuations in zonal and meridional wind components with equal vertical wavelengths but different phases and amplitudes, which is described by Eqs. 1. The left panel of Fig. 1 shows the $u'$ and $v'$ wind fluctuations as a function of altitude. One can

see several oscillations centered around ∼40 km altitude. If we select one full wave period around the center altitude, i.e. from $z_0 - \lambda_z/2$ to $z_0 + \lambda_z/2$ of the QM GW, and plot $u'$ versus $v'$ we get an ellipse as shown in the right panel of Fig. 1. The selected height range with one wave period is marked in Fig. 1a by the shaded area. The major axis of the ellipse is oriented along the wave propagation direction.





For mid- and low-frequency GW the velocity perturbations in propagation direction and perpendicular to this direction are related by the polarization relation (e.g, Gavrilov et al., 1996; Fritts and Alexander, 2003; Holton, 2004):

%beginequation

$$\widehat{v}_\perp = -i(f/\widehat{\omega})\widehat{u}_\parallel \qquad (2)$$

where $\widehat{v}_\perp$ is complex amplitude of wind fluctuations in the direction perpendicular to the direction of propagation and $\widehat{u}_\parallel$ is amplitude of wind fluctuations along the propagation direction. $f = 2\Omega sin\Phi$ is coriolis parameter and $\widehat{\omega}$ is intrinsic frequency. That is, for a zonally propagating wave $\widehat{v}_\perp$ is the meridional velocity amplitude.

The vertical propagation direction of the wave is unambiguously determined by the rotation direction of the zonal wind versus meridional wind hodograph. In the northern hemisphere the (anti-) clockwise rotation of the hodograph indicates a (downward) upward propagating wave.

An additional hodograph of the parallel wind fluctuations versus temperature fluctuations is used to resolve an ambiguity in horizontal propagation direction that arises from the orientation of the ellipse in Fig. 1b. If we assume, that $\widehat{\theta}/\overline{\theta} = \widehat{T}/T_0$ (Fritts and Rastogi, 1985; Eckermann et al., 1998), the temperature amplitude is related to the parallel wind amplitude for a wave propagating in zonal direction as (e.g, Hu et al., 2002; Geller and Gong, 2010):

$$\widehat{T} = \frac{imT_0}{g}\frac{\widehat{\omega}^2 - f^2}{\widehat{\omega}k_h}\cdot\widehat{u}_\parallel = \frac{iT_0}{g}\frac{\sqrt{\widehat{\omega}^2 - f^2}}{\widehat{\omega}}\sqrt{N^2 - \widehat{\omega}^2}\cdot\widehat{u}_\parallel \qquad (3)$$

where $k_h = 2\pi/\lambda_h$ is the horizontal wave number and $\lambda_h$ is the horizontal wavelength of the QM wave. $\widehat{\theta}/\overline{\theta}$ are potential temperature perturbations. $T_0$ and $g$ are the background temperature and the acceleration due to gravity averaged over the altitude range of the QM GW.

Thus, an analysis of the measured fluctuations either using Eqs. 1 and A4 or by means of hodograph technique yields wave characteristics like vertical wavelength, horizontal and vertical propagation direction, and the intrinsic frequency. From these basic parameters we derive further characteristics of the observed GW. This derivation requires precise knowledge of the background, which is obtained from the same measurements. Apart from the fluctuations our advanced lidar measurements yield the mean temperature ($T_0$), buoyancy frequency ($N$) and the absolute wind speed in the altitude range $[z_0 \mp \lambda_z/2]$. With this set of parameters further wave parameters are estimated from equations summarized in Appendix A.

## 4   Retrieval algorithm

In this section we describe the procedure to derive wave parameters from the measured lidar data. For our analysis we need simultaneously measured wind and temperature profiles. Technically we can extract wave parameters from a single measurement, that is using two wind and one temperature profile. However, for a robust estimation of the atmospheric background we need a several hours long observational data set.



### 4.1 Separation of GW and background

The first step is to remove the background from the measured data. The background removal procedure may play a key role in GW-analysis techniques and even lead to strongly biased results. The main reason for this is that the most analysis techniques rely on fluctuation's amplitudes remaining after subtraction of background to infer wave energy (e.g., Rauthe et al., 2008;
Ehard et al., 2015; Baumgarten et al., 2017; Cai et al., 2017, and others). Since GW energy is proportional to amplitude squared, any uncertainty in the background definition ultimately leads to large biases in estimation of GW-energy.

We define the background as wind or temperature fluctuations with periods longer than 12 hours and vertical wavelengths longer than 15 km. To extract such a background from measurements we apply a low-pass filter to the altitude vs time data. Specifically, we use the two dimensional fast Fourier transform (2D-FFT) (e.g., González and Woods, 2002) and, after blocking
the specified high frequencies and short wavelengths, and applying the inverse 2D-FFT, we finally construct the background. Advantage of this method is that it simultaneously accounts for both variability in space and time. After subtracting the derived background from the original measurements we obtain the wind and temperature fluctuations which might only be produced by gravity waves. This procedure is demonstrated in Fig. 2, 3, and 4 for temperature, zonal, and meridional wind, respectively. The lower panels represent the $T'$, $u'$, $v'$ fluctuations that are analyzed with our automated hodograph method. These time-altitude
plots consist of many single-time ("instant") altitude-profiles which are further analyzed individually.

We also made a robustness test to check how different background removals influence our advanced hodograph-based method. To derive the background (for both wind and temperature data) we made use of (a) running mean with different smoothing window lengths, (b) different splines, and (c) constant values in time. It turned out that our analysis results were near identical for all these different backgrounds. The new technique is not sensitive to the background derivation schemes and
may even allow to skip this step from the analysis. A more in depth analysis showed, that the robustness to the background removal is a consequence of the analysis approach. We only search for waves which are prominent simultaneously in temperature and both wind components. Even though we are confident in the robustness of our technique to the various background derivation methods, we consider the 2D-FFT based approach as most adequate for this purpose.

### 4.2 Scaling of fluctuations

Without wave breaking and dissipations, the amplitude of fluctuations increases with altitude as $\exp(z/(2H))$. In the real observations, since waves cannot freely propagate throughout the atmosphere, the amplitude of the fluctuations increases with altitude as $\exp(z/(\varsigma H))$, where the coefficient $\varsigma \geq 2$ is derived from the observed data. The exponential growth, however also affects any analysis, in particular wavelet analysis, since normalization is always applied. The growing amplitude works as a weighting function and, thereby prohibits analysis of small-scale features at lower altitudes. Scaling the fluctuations by
$1.0/\exp(z/(\varsigma H))$ yields fluctuations with comparable amplitudes over the whole altitude range. For the observations presented here we use $\varsigma = 2.15$.



### 4.3 Detection of wave packets

Starting from this point we only analyze the altitude-profiles at every time step. At every time step we have measured profiles of wind and temperature which are split in fluctuations and background profiles.

First, we search for dominant waves in both altitude and wave number domains. For this purpose we apply the continuous wavelet transform (CWT) to every profile of the extracted fluctuations. We use a Morlet wavelet of the sixth order (Torrence and Compo, 1998) and apply it to vertical profiles of wind and temperature fluctuations. Similar procedure was also applied by Zink and Vincent (2001) and Murphy et al. (2014). By applying wavelet analysis they define regions from which the Stokes analysis (e.g., Vincent and Fritts, 1987; Eckermann, 1996) is further evaluated with a better precision. We note here, that their results rely on accuracy of wavelet transform and on assumption that wave signatures are well separated from each other and clearly resolved by CWT.

An example for the resulting scalograms of one time step is shown in Fig. 5. These scalograms are normalized to unity to make spectral signatures comparable between the different fluctuations. In zonal wind and temperature fluctuations a clear peak between $\sim$40 and $\sim$55 km with a vertical wavelength of approximately 10 to 15 km can be seen. Both wind components reveal peaks below $\sim$40 and above $\sim$60 km with wavelengths of about 5 km. As a next step we combine these wavelet spectra and construct a single scalogram that reflects the features common for all three components. We calculate the product of all three spectra and define this as the combined spectrum. Note, that Zink and Vincent (2001) and Murphy et al. (2014) used sum of scalograms of both wind components. The combined scalogram in Fig. 6 reveals one large (around 10 km wavelength) and two smaller (near 35 and 70 km altitude) regions with weaker wave amplitudes. The larger region is relatively broad and reveals a vertical wavelength increase with increasing altitude. This can be due to two reasons: 1) it is one wave packet with changing vertical wavelength due to variable background or 2) it is a sum of two wave packets with overlap at around 50 km altitude. This uncertainty is difficult to resolve just using information from wavelet transform. To resolve this ambiguity we developed a sequence of further analysis steps and only use these CWT results as an input (zero guess) for further analysis.

### 4.4 Fitting of linear wave theory

We start with the larger area encircled by the dashed lines in Fig. 6 and fit the Eqs. 1 to the wind and temperature profiles. Note, that Eqs. 1 include scaling factor $\exp(z/(2H))$.

However, after applying step 4.2 of our algorithm, we get rid of exponential growth in fluctuations profiles and, thereby exclude this factor from wave equations. Thus for fitting single profiles we use the remaining functions:

$$T' = |\widehat{T}| \cdot \exp(-(z-z_0)^2/2\sigma^2) \cdot \cos(m(z-z_0) + \varphi_T) \tag{4a}$$

$$u' = |\widehat{u}| \cdot \exp(-(z-z_0)^2/2\sigma^2) \cdot \cos(m(z-z_0) + \varphi_u) \tag{4b}$$

$$v' = |\widehat{v}| \cdot \exp(-(z-z_0)^2/2\sigma^2) \cdot \cos(m(z-z_0) + \varphi_v) \tag{4c}$$





We recall that the introduced in Sec. 3 vertical extend of wave packet $\exp(-(z-z_0)^2/2\sigma^2)$ is essential for analysis of observations which cover long, $\sim 50\ km$ altitude range.

The results obtained from the wavelet transform in the altitude range ($z_0 \simeq 45$ km, $\lambda_z \simeq 12$ km) are used as a zero guess. The obtained fitting results yield intrinsic frequency and propagation direction (c.f. Eq. A4). However, by testing different

simulated and measured data we concluded that for GW with intrinsic periods larger than $\sim$1 hour the hodograph analysis yields more accurate results than those based on the fitting of Eqs. 4. Thus, the fitting procedure is only used to precisely derive the altitude $z_0$ and the vertical wavelength $\lambda_z = 2\pi/m$ of the wave packet, which are smeared in the spectrogram (Fig. 6), and continue our analysis using the hodograph technique. The updated values for this case are $z_0 = 49$ km, $\lambda_z = 11$ km.

## 4.5    Hodograph method

According to the theory described in Sec. 3 and App. A the $u'$ and $v'$ fluctuations form an ellipse if the intrinsic period is between $\sim$1 and $\sim$12 hours. For Higher frequency GW, i.e. with periods below $\sim$1 hour the fluctuations form a line, as the influence of the Coriolis force is negligible. For low frequency GW, i.e. those with periods close to the Coriolis period ($2\pi/f$) the fluctuations reveal a circle.

In order to minimize an error in the hodograph analysis due to presence of other waves (Zhang et al., 2004), we apply a

vertical band-pass filter to all 3 profiles and thereby remove waves with wavelengths shorter than $\lambda_z/2$ and longer than $2\lambda_z$. For example, if the vertical wavelength obtained in the previous step is 10 km, we remove waves with wavelengths shorter (longer) than 5 (20) km. Using a two dimensional least square fitting software we find the best fit parameters that satisfy the ellipse equation (Fitzgibbon et al., 1996). The fitting procedure is sensitive to the data quality and if for example the data is far away from an elliptical shape, the fitting procedure does not converge. Only if the ellipse was successfully fitted, we extract

further wave characteristics from this data set.

## 4.6    Optimization of results

The rotation direction of the hodograph is defined as a phase angle change of either $u'$ or $v'$ from the bottom level to the top level over the height region $[z_0 - \lambda_z/2,\ z_0 + \lambda_z/2]$.

As described in sec. 3, the rotation direction indicates whether the wave propagates upwards (in case of clockwise rotation)

or downwards (counterclockwise rotation). Furthermore, if the rotation does not make a full 360 ° cycle, this suggests either an inconsistency in the hodograph results (sec. 4.5) and the wave fit (sec. 4.4) or the vertical extent of the wave packet is smaller than vertical wavelength. Additionally we calculate a vertical wavelength by requiring the hodograph to close the full 360 ° cycle. In the case that the new (corrected) wavelength $\lambda_z$ differs significantly from $\lambda_z$ obtained before (sec. 4.4), we repeat the hodograph analysis using the new (corrected) wavelength.





## 4.7 Calculation of GW parameters

The ratio of the major and minor ellipse axes is further used to derive the intrinsic frequency of GW (App. A). The analysis presented so far allows to derive the intrinsic wave frequency, vertical wavelength, and up- or downward propagation. The horizontal direction of propagation is along the major axis of the ellipse with a remaining uncertainty of 180 °. To resolve this
ambiguity we further use temperature fluctuations profile as described in sec. 3. Specifically, we construct a hodograph from temperature fluctuations and wind fluctuations along the wave propagation direction, i.e. parallel to the derived wave vector. The rotation direction of this new hodograph finally defines the direction of the wave vector: for upward propagating GW clockwise or counterclockwise rotation indicates eastward or westward direction, respectively. For downward propagating GW opposite direction has to be used (Hu et al., 2002).
Knowing all these wave parameters and applying linear wave theory we derive further wave characteristics as described in App. A. Fig. 7(a-c) show the fluctuations and two hodographs defined from the two maxima shown in Fig. 6. Results obtained from this example are summarizes in Table 1 (first column).

## 4.8 Iteration process

After the first QM GW is identified in all three profiles, it is subtracted from those data. We repeat the procedure described
above for all of the maxima seen in the combined spectrogram (Fig. 6). That is, the dominating frequency is used as a zero guess for the fitting of Eqs. 1 to derive exact values of $z_0$ and $\lambda_z$ and then the hodograph analysis is applied to derive the extended set of GW parameters (from 4.3 to 4.7 ). In order to avoid over fitting, we limit our analysis to maximum of 20 waves per one time step. As it will be demonstrated in the next section that we never reach this limit. In the given example 5 waves were detected. The first 3 waves are demonstrated in Fig. 7 and the obtained parameters are summarizes in Table 1. In this example we found
that a wave with a vertical wavelength of 16.4 km propagates upward and against background wind in the altitude range from 56 to 73 km. In the altitude range of 44 to 54 km a wave with 11 km vertical wavelength propagates downward and with nearly the same direction as background wind. The analysis indicates that the broad maximum in the combined spectrogram (Fig. 6) was produced by the sum of two wave packets with different characteristics.
    Finally, this algorithm for a single point in time is subsequently applied to all time points of the entire data set shown in
Fig. 2, 3 and 4.



**Table 1.** Examples of hodograph results from 10 Jan 2016 02:07:30

|  | wave 1 | wave 2 | wave 3 |
|---|---|---|---|
| vert. propagation | downward | upward | upward |
| altitudes $(km)$ | 44 - 54 | 30 - 34 | 56 - 73 |
| vertical wavelength $\lambda_z$ $(km)$ | 11 | -4.8 | -16.4 |
| major axis of the ellipse $\widehat{u}_\parallel$ $(m/s)$ | 12.41 | 9.3 | 17 |
| minor axis of the ellipse $\widehat{v}_\perp$ $(m/s)$ | 2.25 | 4.8 | 4 |
| horizontal propagation angle | 23 | 235.6 | 182 |
| horizontal propagation angle from Eq. A4 | 21 | 233 | 189 |
| ratio of major to minor axis of the ellipse $\widehat{u}_\parallel/\widehat{v}_\perp$ | 5.53 | 1.93 | 4 |
| intrinsic period $(h)$ | 2.3 | 6.64 | 3 |
| horizontal wavelength $\lambda_h$ $(km)$ | 279 | 530 | 513 |
| intrinsic phase speed $(m/s)$ | 33.5 | 22.2 | 46 |
| background zonal wind speed $u_0$ $(m/s)$ | 94.75 | 44.7 | 40.6 |
| background meridional wind speed $v_0$ $(m/s)$ | 5 | 1.7 | 6.35 |
| wind magnitude $\sqrt{u_0^2 + v_0^2}$ $(m/s)$ | 95 | 44.72 | 41 |
| wind magnitude along wave propagation $(m/s)$ | 89.3 | 24 | 40 |
| observed period $(h)$ | 0.63 | -80.5 | 19 |
| temperature $(K)$ | 270.5 | 233 | 265 |
| buoyancy frequency $(1/s)$ | 0.019 | 0.025 | 0.0172 |
| kinetic energy $(J/kg)$ | 53.2 | 27.8 | 71 |
| potential energy $(J/kg)$ | 50 | 15 | 64 |
| vertical flux of horizontal pseudomomentum $(m^2/s^2)$ | 3 | 0.4 | 4.6 |





## 5 Results and discussion

In this section we demonstrate on a real data set how our analysis works and results are summarized in form of different statistics.

The data used in this study were obtained from 09 to 12 January 2016. During this time period a strong jet with wind speeds of more than 100 m/s was observed at an altitude range of 45 to 55 km (Fig. 3 and Fig. 4). During this period maps of the horizontal winds extracted from ECMWF-IFS (European Centre for Medium-Range Weather Forecasts - Integrated Forecasting System) showed a strong polar Vortex with wind speeds of more than 160 m/s at the vortex edge. The Vortex was elongated towards Canada and Siberia and its center displaced towards Europe. ALOMAR was located roughly below the Vortex Edge where the Polar Night Jet was located south of ALOMAR, at about 60°N with wind speeds of more than 160 m/s.

After applying the new analysis technique to the ∼60 hours measurements shown in Figs. 2, 3, and 4 we obtain the following results.

The number of detected waves per altitude profile is summarized in histogram Fig. 8. In 645 out of 715 altitude profiles we find at least one height range with a dominant GW where the hodograph analysis provides a reliable result. We recall that the analysis technique allows for up to 20 waves in a single profile. The total number of the detected waves amounts to 4507. It is seen that the majority of profiles yields 5 to 10 waves and none of them reaches the 20-waves limit. From the rotation direction of the velocity hodographs we derive that 32.3 % of all the detected waves propagate downwards.

This finding can only qualitatively be compared with other measurements, since most of them were done at different altitudes or latitudes. Hu et al. (2002) found 223 (71 %) waves propagating upwards and 91 (29 %) downwards in the altitude range 84–104 km, which is in accord with our results. Gavrilov et al. (1996) reported that up to 50 % of the detected waves propagate downwards in the altitude range 70 to 80 km. In the troposphere and lower stratosphere (below 20 km) Sato (1994) reported less than 10 % downward propagating GW and Mihalikova et al. (2016) reported 18.4 % during wintertime and 10.7 % during summertime. From rocket observations of zonal and meridional wind components with a vertical resolution of 1 km in altitude range 30 to 60 km Hirota and Niki (1985) found in middle and high latitudes about 20 % of downward propagating GW, and 30–40 % in low latitudes at northern hemisphere stations. At the only southern hemisphere station (Ascension Island) 36 % of downward propagating GW were observed (Hirota and Niki, 1985). Hamilton (1991) found from rocketsonde observations of wind and temperature in the 28–57 km height range at 12 stations (spanning 8°S to 76°N) different fractions of downward propagating GW spanning from 2 % to 46 % depending on latitude and season. Wang et al. (2005) reported that approximately 50 % of the tropospheric gravity waves show upward energy propagation, whereas there is about 75 % upward energy propagation in the lower stratosphere. From their radiosonde observations authors demonstrate that the lower-stratospheric fraction of upward energy propagation is generally smaller in winter than in summer, especially at mid- and high latitudes. Thus, our finding of 32.3 % downward propagating GW reasonably agrees with other experimental data. We note, that the observed downward or upward propagating GW are instantaneous observations, which means that we have no information about the fate of the observed waves. I.e., we cannot estimate the percentage of waves which ultimately get to the ground.





Fig. 9 shows details of the wave packets as functions of altitude and separated for upward and downward propagating GW. First plot shows the number of wave center altitudes ($z_0$) and does not consider the vertical extent of the wave packets (vertical wavelengths). The latter is taken into account in the middle panel which shows the mean fraction of the profile where a wave packet is present (any part of the wave, center or tail). We find that the most active regions (in terms of number of GW) are ∼32 to 40 km and 58–64 km. The altitude region between ∼40 and 55 km contains the smallest number of the detected waves.

It is interesting to compare these results with mean background wind shown in the rightmost panel of Fig. 9. It is obvious that the minimum in the wave activity as deduced by our analysis technique is co-located with the maximum of mean zonal wind as well as the background temperature.

To investigate the time and altitude dependence of the GW detected by our hodograph technique, we reconstructed the temperature and the wind fluctuation fields from the derived waves parameters using Eqs. 1. Fig. 10 shows the result of this reconstruction for the temperature fluctuations separated for upward and downward propagating GW. Contour lines show the background zonal wind velocity. We recall that the analysis technique treats every single altitude-profile independently and, therefore the influence of neighboring profiles is only due to time averaging. It is therefore remarkable, that the joint field of reconstructed GW shown in Fig. 10 builds up a consistent picture. Thus one can recognize for instance, waves packets of several hours duration. In some cases phase lines of waves follow the background wind. For example on 11 January after 18:00 UT at altitudes between 54 and 63 km a maximum of temperature fluctuations of upward propagating waves follows the contour line of a zonal wind of 60 m/s.

We use similar representations to investigate the temporal variability of any other of the derived GW properties. For example Fig. 11 summarizes the obtained intrinsic periods of GW throughout the measurement. On the one hand these figures demonstrate high variability, but on the other hand they also show regions of consistent picture. For example, on 11 January after 21 UT at altitudes between 54 and 62 km one can see wave period of about 7 hours for ∼ 2 hours. The analysis allows studying the temporal and altitude variation of the wave periods, e.g. upward propagating low period waves with large vertical wavelengths are often found above the jet maximum.

In Fig. 12 we show distributions of the derived GW parameters for all identified waves. One remarkable feature seen in these histograms is that the distributions of wavelengths and phase velocities reveal very similar shapes for up- and downward propagating waves. The distributions of intrinsic periods show quite different kurtosis for up- and downward propagating GW. These histograms also demonstrate limitations of the presented analysis. Only a few waves with intrinsic periods smaller than 1 h or with vertical wavelengths below 1 km are detected. This is likely caused by the smoothing of the lidar data with a Gaussian window of 15 minutes and 0.5 km rather than to the hodograph method itself. Waves with vertical wavelengths above ∼15 km were likely associated to background fluctuations when applying the 2D-FFT.

The distribution of phase velocities in Fig. 12 demonstrates that the velocities are below 60 m/s with a maximum of occurrence at ∼10 m/s. Matsuda et al. (2014) estimated horizontal GW phase velocities from airglow images. Their waves had periods below ∼1 hour and revealed phase speeds between 0 and 150 m/s. Among those waves, ∼70 % showed phase speeds between 0 and 60 m/s. In our case, the observed waves periods have maximum in the range 4 to 5 hours and, as expected from Eq. A10, the horizontal phase velocity is also lower than those, reported by Matsuda et al. (2014).



Next, we analyze and sum up the wave energetics. Fig. 13 shows the derived kinetic ($E_{\mathrm{kin}}$) and potential ($E_{\mathrm{pot}}$) energy densities as well as their statistical basis. The altitude dependence of the energy distribution is shown as color coded 2-d histogram (color bar on the right hand side defines number of waves). For these histograms all waves were used, i.e. both propagating up- and downwards. Fig. 13 also shows the mean energies separated for up and downward propagating waves. We

find that $E_{\mathrm{kin}}$ of downward propagating GW is lower than $E_{\mathrm{kin}}$ of upward propagating waves and $E_{\mathrm{pot}}$ is nearly identical for up- and downward propagating GW. The standard method to derive $E_{\mathrm{kin}}$ and $E_{\mathrm{pot}}$ from ground based observations is to average bulk wind and temperature fluctuations and apply Equations A13 and A15. Fig. 13 shows that the fluctuation-based method reveals a good agreement with mean profiles derived from our new retrievals. These results are also in agreement with mean winter profiles measured at ALOMAR observatory, summarized by Hildebrand et al. (2017).

The directions of background wind and wave propagation are summarized in Fig. 14 as polar histograms. The leftmost part, i.e. Fig. 14a shows a histogram of the background wind at the time/altitude of every hodograph. The analysis shows that in almost all cases the wind in the vicinity of detected waves blows towards the east-north-east with a mean speed of about 70 m/s.

Figs. 14b and 14c show polar histograms of the detected upward and downward propagating waves. From the color code

we see, that the horizontal phase speed of the upward propagating waves is in general larger than that one of the downward propagating waves. Downward propagating waves reveal a rather uniform spatial distribution whereas upward propagating waves prefer to propagate against the background wind.

To address the question at which vertical angles the GW propagate we show histograms of the angle between the group velocity vector and the horizon ($\beta$; Eq. A12) separated for up- and downward propagating waves in Fig. 15. In the beginning of

this section we noted that our analysis reveals that $\sim 30\,\%$ of all the detected waves propagate downwards. From the histograms we note that this difference of up- and downward propagating waves is mostly due to waves propagating at shallow angles of less than $\sim 1$ degree. GW with larger vertical angles are found in same numbers for upward and downward propagating waves.

The vertical group velocities $c_{gz}$ estimated using Eq. A11 are summarized in Fig. 16 for up- and downward propagating waves. Since the vertical group velocity depends on wave periods, we split the histograms in two groups of longer and shorter

than 8 hours. These results show for instance, that all low frequency waves (in the range of frequencies considered in our study) reveal small vertical group velocities. Waves with periods shorter than 8 hours show a somewhat more complicated picture. The vertical group velocities of downward propagating waves exceed those of upward propagating GW for waves that propagate in the direction of the background wind. In turn, upward propagating GW reveal highest vertical group velocities if they propagate against the background wind. The vertical group velocities $c_{gz}$ are at least two times lower than vertical phase

speeds ($c_z$, not shown here). The values of the vertical group velocity imply, that if waves propagate from the ground to the altitude where they were observed, they need 6 to 14 (2 to 4) days if they have period longer (shorter) than 8 hours. Somewhat similar time scales for GW to reach the lower stratosphere were reported by Sato et al. (1997) whose group velocity for waves with period of 17 h were 1.7 km/day.

Finally we show vertical fluxes of horizontal momentum (see Eq. A18) averaged over periods from 2 to 12 hours, which is

a key quantity for atmospheric coupling by waves in Fig. 17. This plot demonstrates, that for these measurements the vertical




flux of horizontal momentum rapidly decreases with altitude up to ∼45 km. Above ∼42 km it remains rather constant up to ∼70 km. In the altitude range from 42 to 70 km where we find a low variability of the momentum flux we analyzed its dependence on the horizontal propagation direction of the waves. The result is shown as polar histograms in Fig. 18. We see that the momentum flux of downward propagating waves is lower than that of upward propagating GW. Fig. 18 also shows that

waves propagating nearly perpendicular to the mean wind carry the smallest flux for both up- and downward propagating GW. Note, that the direction of the momentum flux is not necessarily along the major axis of the ellipse. The angle between the directions of momentum flux and GW propagation was estimated by Eq. 13 from Gavrilov et al. (1996) and does not exceed 2.8°, which is much lower than the width of the bins in the histograms.

## 6  Summary and conclusion

In this paper a detailed step-by-step description of a new algorithm for derivation of GW parameters with justification of every step is presented. Most of these steps if considered independently, are well known and validated in numerous experimental works. The advantage and novelty of this work is their combination and some justifications on their importance and how they affect GW-analysis results.

Thus e.g., very first action normally performed on the measured time series is background removal. Since most conventional

techniques based on smoothing or averaging in time or altitude ultimately introduce artifacts, we introduced a new method, namely applying 2D-FFT for the background removal. Advantage of this method is that it simultaneously accounts for both variability in space and time. Moreover, to avoid any degree of arbitrariness, the background removal can be excluded from fluctuation analysis when applying further steps of the analysis technique described in the manuscript.

As a next step we proposed to apply a scaling function of the form $\exp(z/(\varsigma H))$, where $H$ is scale height, $z$ is altitude, and

the constant $\varsigma$ can be derived by a linear fit to fluctuation profiles and should be in a range 1–10 (we derived $\varsigma = 2.15$ for our data). This, to our knowledge, a new technique which is not explicitly described in the literature. Advantage of this approach is to suppress exponential growth of GW-amplitudes to allow for equally weighted detection of wave signatures within the entire altitude range. This e.g., is clearly seen in a wavelet scalograms which would otherwise be predominantly sensitive to strongest amplitudes, hiding out waves at lower altitudes.

The most essential part of the proposed analysis technique consist of fitting of cosines-waves to simultaneously measured profiles of winds and temperature and subsequent hodograph analysis of these fitted waves. We emphasize that this fit must be applied to all three quantities, i.e. zonal and meridional wind and temperature (u, v, and T), simultaneously. This ensures that we deal with a real GW which leaves its signature in all these physical quantities that were measured simultaneously in the same volume. The main difficulty in application of the hodograph analysis to real measurements is to find the wavelengths and

altitude regions where certain GW dominates all measured quantities (u, v, and T). Since very often the measured data represent a mixture of vast different GWs, it is generally very difficult to find them automatically in the frame of hodograph analysis. Thus, this work was always accomplished manually, by applying visual check of data and analysis quality. So was also done in particular by Baumgarten et al. (2015). The novelty of our approach is that its robustness ultimately allows for automation of





the hodograph analysis. Also, our algorithm resolves many more GWs than it can be inferred by manually applied hodograph technique.

All these advantages are especially important since modern advanced measurement techniques (e.g. our lidar system described in Sec. 4) are capable of doing long duration measurements that cover large altitude range ∼30 to 80 km. This huge amount of data requires a robust and stable automatic analysis technique which we developed and presented in this work.

One obvious advantage of the proposed algorithm is that it allows for simultaneous detection of any kind of waves presented in the measurements. This includes not only GWs, but also tides. That is, the necessity of removal tidal components a priori, which cannot be done unambiguously, is eliminated. All the detected waves can be sorted out on a statistical bases after the observational data base is analyzed by using the proposed algorithm.

At the current stage of development our analysis technique, however, is not capable of detecting life-time of gravity waves in observational data set. This capability is currently under development as well as an additional robust algorithm to pick out wave packets automatically.

By applying this new methodology to real data obtained by lidar during about 60 hours of observations in January 2016 we found 4507 single holographs. In general, 5 to 10 waves were detected from every vertical profile. This allowed identifying and analyzing quasi monochromatic waves in about ∼80 % of the observations. The measurements were performed while a jet at the stratopause (45–55 km) of more than 100 m/s was located above the lidar station. We found a strong decrease in vertical flux of horizontal momentum up to ∼42 km altitude. Due to the strong wind above ∼40 km, it is likely that waves break, get absorbed, and reflected below this altitude region. The new method allows studying waves separated for up- and downward propagation according to their group velocities.

The main characteristics of upward and downward propagating GW were investigated statistically. We find that downward propagating GW reveal shorter intrinsic periods and slower phase speeds than upward propagating GW. Downward waves propagate at steeper angles than the upward propagating ones. Currently our analysis does not allow to distinguish between primary and secondary GW. The next step will be to look for similar wave characteristics (horizontal, vertical wavelengths, and propagation direction) in upward and downward propagating waves. The nearby occurrence of similar waves with opposite vertical propagation direction is an indication of secondary GW.

https://doi.org/10.5194/amt-2019-79
Preprint. Discussion started: 27 June 2019

Atmospheric Measurement Techniques Discussions

**Appendix A: Theoretical basis and formulary**

A monochromatic gravity wave (GW) perturbation in Cartesian coordinates (x, y, z) with wave number components ($k$, $l$, $m$) and ground relative (Eulerian) frequency $\omega$ can be written in the following form (e.g, Gill, 1982; Fritts and Alexander, 2003; Holton, 2004):

$$T' = Re\{\widehat{T} \cdot exp(i(kx + ly + mz - \omega t))\} \cdot exp(z/2H) \quad \text{(A1a)}$$

$$u' = Re\{\widehat{u} \cdot exp(i(kx + ly + mz - \omega t))\} \cdot exp(z/2H) \quad \text{(A1b)}$$

$$v' = Re\{\widehat{v} \cdot exp(i(kx + ly + mz - \omega t))\} \cdot exp(z/2H) \quad \text{(A1c)}$$

where $\widehat{T}$, $\widehat{u}$ and $\widehat{v}$ are complex amplitudes of temperature, zonal and meridional wind fluctuations and $H$ is scale height. Alternatively, these equations can be rewritten in form:

$$T' = |\widehat{T}| \cdot cos(kx + ly + mz - \omega t + \varphi_{T0}) \cdot exp(z/2H) = |\widehat{T}| \cdot cos(mz + \varphi_T) \cdot exp(z/2H) \quad \text{(A2a)}$$

$$u' = |\widehat{u}| \cdot cos(kx + ly + mz - \omega t + \varphi_{u0}) \cdot exp(z/2H) = |\widehat{u}| \cdot cos(mz + \varphi_u) \cdot exp(z/2H) \quad \text{(A2b)}$$

$$v' = |\widehat{v}| \cdot cos(kx + ly + mz - \omega t + \varphi_{v0}) \cdot exp(z/2H) = |\widehat{v}| \cdot cos(mz + \varphi_v) \cdot exp(z/2H) \quad \text{(A2c)}$$

where general phase shift in form of $\varphi_i = kx + ly - \omega t + \varphi_{i0}$ (subscript $i$ refers to either of $T, u$ or $v$) was introduced. For observations of one vertical profile, the quantity $(kx + ly - \omega t)$ contributes to the fluctuations as a phase shift.

Finally, we take into account that quasi monochromatic (QM) gravity wave (GW) is limited in space, i.e. appears in our observations within a limited altitude range:

$$T' = |\widehat{T}| \cdot exp(-(z - z_0)^2/2\sigma^2) \cdot cos(m(z - z_0) + \varphi_T) \cdot exp(z/2H) \quad \text{(A3a)}$$

$$u' = |\widehat{u}| \cdot exp(-(z - z_0)^2/2\sigma^2) \cdot cos(m(z - z_0) + \varphi_u) \cdot exp(z/2H) \quad \text{(A3b)}$$

$$v' = |\widehat{v}| \cdot exp(-(z - z_0)^2/2\sigma^2) \cdot cos(m(z - z_0) + \varphi_v) \cdot exp(z/2H) \quad \text{(A3c)}$$

where $\sigma$ is a factor, describing width of wave packet , $z_0$ altitude of maximum wave amplitude.
Following Cot and Barat (1986); Gavrilov et al. (1996), the horizontal propagation angle of QM GW can be defined as follows:

$$\xi = \frac{1}{2}(\pi n + \arctan(\frac{2\Phi_{uv}}{\widehat{v}^2 - \widehat{u}^2})) \quad \text{(A4)}$$

where $\xi$ is the azimuth angle of wave propagation direction and $\Phi_{uv} = \widehat{u} \cdot \widehat{v} \cdot cos(\varphi_u - \varphi_v)$. The integer $n = 1$ when $\widehat{v} < \widehat{u}$. When $\widehat{v} > \widehat{u}$, $n = 0$ and 2 for $F_{uv} > 0$ and $F_{uv} < 0$, respectively. This implies, that for $\varphi_u - \varphi_v = \pi/2$ propagation direction can be 0 or 180 degrees, i.e. northward or southward if $\widehat{v} > \widehat{u}$ and eastward or westward if $\widehat{v} < \widehat{u}$. The sign of $m$ in





Eqs. 1 shows the vertical propagation direction: $m < 0$ for upward and $m > 0$ for downward propagating GW. This theoretical basis allows to describe the main GW-parameters and to derive them from observations. However in practice, noisy data and/or insufficient resolution of measurements may lead to large uncertainties when applying these equations directly to the measured time series.

Most common technique, based on linear theory of gravity waves to derive propagation direction, intrinsic frequency and phase velocity of GW from ground-based observations is the hodograph method (e.g., Cot and Barat, 1986; Sawyer, 1961; Wang and Geller, 2003; Zhang et al., 2004; Baumgarten et al., 2015).

    In order to keep polarization relation as simple as Eq. 2, we can rotate coordinate system $(x, y)$ with wave wind fluctuations $u'$ and $v'$ to $(x_\parallel, y_\perp)$ -Cartesian coordinate system in which the origin is kept fixed and the $x_\parallel$ and $y_\perp$ axes are obtained by

rotating the $x$ and $y$ counterclockwise through an angle $\pi/2 - \xi$. In new coordinate system wave propagate along $x_l$ axe and amplitudes ratio in new coordinate system is:

$$|\widehat{u_\parallel}|/|\widehat{u_\perp}| = \widehat{\omega}/f \tag{A5}$$

Relationship between fluctuations in new $(x_\parallel, y_\perp)$ and standard coordinate systems $(x, y)$ are:

$$u' = u'_\parallel \cdot sin(\xi) - v'_\perp \cdot cos(\xi) \tag{A6a}$$
$$v' = u'_\parallel \cdot cos(\xi) + v'_\perp \cdot sin(\xi) \tag{A6b}$$
$$u'_\parallel = u' \cdot sin(\xi) + v' \cdot cos(\xi) \tag{A6c}$$
$$u'_\perp = -u' \cdot cos(\xi) + v' \cdot sin(\xi) \tag{A6d}$$

Amplitudes of ellipse in new coordinate system Gavrilov et al. (1996) are:

$$2\widehat{u_\parallel}^2 = \widehat{u}^2 + \widehat{v}^2 + \sqrt{(\widehat{u}^2 - \widehat{v}^2)^2 + 4\Phi_{uv}^2} \tag{A7a}$$
$$2\widehat{u_\perp}^2 = \widehat{u}^2 + \widehat{v}^2 - \sqrt{(\widehat{u}^2 - \widehat{v}^2)^2 + 4\Phi_{uv}^2} \tag{A7b}$$

    Thus, $\widehat{u_\parallel}$ and $\widehat{u_\perp}$ can be derived from fitting of ellipse to wind vector or by fitting Eqs. A3 to the data and applying Eqs. A7. Afterwards Eq. A5 is used to derive intrinsic frequency $\widehat{\omega}$ of the wave.

On the other hand the intrinsic frequency is a function of buoyancy frequency (N), coriolis parameter $f$ and angle $\alpha$, which is the angle between phase lines and vertical (Holton, 2004, Eq. 7.56) :

$$\widehat{\omega}^2 = N^2 \cos^2 \alpha + f^2 \sin^2 \alpha \tag{A8}$$





From this equation the horizontal wave number along propagation direction can be derived (Fritts and Alexander, 2003; Vaughan and Worthington, 2007):

$$k_\parallel^2 = m^2 \left( \frac{\widehat{\omega}^2 - f^2}{N^2 - \widehat{\omega}^2} \right) \tag{A9}$$

The horizontal/vertical phase speed is the ratio of intrinsic frequency to horizontal/vertical wave number (e.g., Nappo, 2002):

$$c_\parallel = \widehat{\omega}/k_\parallel \tag{A10a}$$

$$c_z = \widehat{\omega}/m \tag{A10b}$$

The vertical component of the group velocity $c_{gz}$ of the hydrostatic inertia gravity waves is given by (Gill, 1982; Sato et al., 1997):

$$c_{gz} \equiv \frac{\partial \widehat{\omega}}{\partial m} = -\frac{(N^2 - f^2)k_\parallel^2 m}{\widehat{\omega}(k_\parallel^2 + m^2)^2} \simeq -\frac{N^2 k_\parallel^2}{\widehat{\omega} m^3} \tag{A11}$$

The angle between the group velocity vector and the horizon can be estimated from $\alpha$ as:

$$\beta = \pi/2 - \alpha \tag{A12}$$

Kinetic energy density of GW estimated from observed fluctuations (e.g., Gill, 1982; Holton, 2004; Placke et al., 2013):

$$E_{kin} = \frac{1}{2}\overline{(v'^2 + u'^2)} \tag{A13}$$

Thus, kinetic energy density as a function of fitted amplitudes of wind hodograph:

$$E_{kin} = \frac{1}{4}\left( \widehat{v}_\parallel^2 + \widehat{u}_\perp^2 \right) \tag{A14}$$

Potential energy density of GW estimated from observed fluctuations (e.g., Holton, 2004; Geller and Gong, 2010; Placke et al., 2013):

$$E_{pot} = \frac{1}{2}\frac{g^2}{N^2}\frac{\overline{T'^2}}{T_0^2} \tag{A15}$$

$E_{pot}$ from amplitudes of temperature fluctuations:

$$E_{pot} = \frac{1}{4}\frac{g^2}{N^2}\frac{\widehat{T}^2}{T_0^2} \tag{A16}$$

Vertical flux of horizontal momentum in wave propagation direction can be written as (e.g., Fritts and Alexander, 2003):

$$F_{P\parallel} = \overline{\rho}\left( 1 - \frac{f^2}{\widehat{\omega}^2} \right)\overline{u_l'w'} \tag{A17}$$





where $w'$ is vertical wind fluctuations and $\overline{\rho}$ is the atmospheric density. From continuity equation we get $w' = -(k_\parallel/m) \cdot u'_l$ and the vertical momentum flux is transformed to (e.g., Réchou et al., 2014):

$$F_{P\parallel} = \frac{\overline{\rho}}{2} \left(1 - \frac{f^2}{\widehat{\omega}^2}\right) \frac{k_\parallel}{m} \, \widehat{u}_l^2 \tag{A18}$$

*Author contributions.* IS developed the analysis technique code and performed the calculations. GB designed experiments and conducted
5   measurements; GB and IS analyzed the data; IS, GB, and FJL contributed to the final manuscript.

*Competing interests.* The authors declare that they have no conflict of interest.

*Data availability.* The data used in this paper are available upon request.

*Acknowledgements.* This study benefited from the excellent support by the dedicated staff at the ALOMAR observatory. The DoRIS project
was supported by Deutsche Forschungsgemeinschaft (DFG, German Research Foundation, no. BA2834/1-1). This project has received
10   funding from the European Union's Horizon 2020 Research and Innovation programme under grant agreement no. 653980 (ARISE2) and was
supported by the Deutsche Forschungsgemeinschaft (DFG, German Research Foundation) under project LU1174/8-1 (PACOG), FOR1898
(MS-GWaves).





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



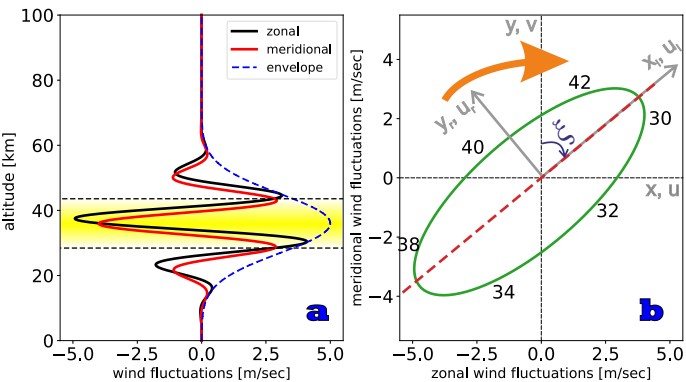

**Figure 1.** Schematics of the method. (a) Altitude profile of horizontal velocity fluctuations. Blue dashed line demonstrates an envelope. Colored area marks altitude range of one wavelength where wave amplitude is most significant ($[z_0 - \lambda_z/2, \; z_0 + \lambda_z/2]$) (b) Hodograph ellipse of IGW horizontal velocity variations taken from altitude range marked in plot (a). Dashed line shows major axis of ellipse, which is a propagation direction of the wave. Numbers around ellipse are altitudes. In this schematics clockwise rotation



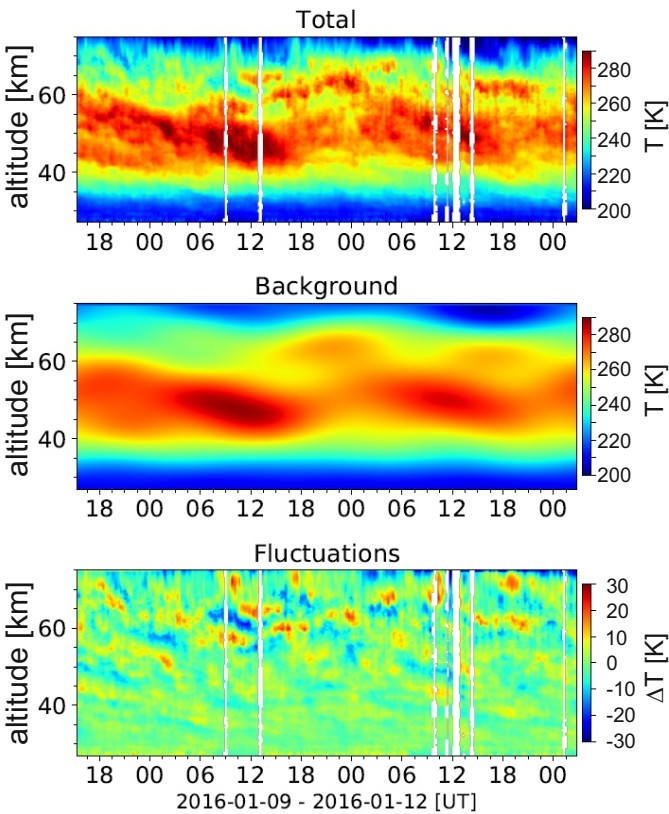

**Figure 2.** Temperature observations. Upper panel demonstrates temperature observations obtained from 9-12 January 2016. In middle panel background obtained by 2D-fft is demonstrated. Lower panel shows remaining small scale fluctuations used for GW analysis. White vertical lines represent gaps in the measured data.

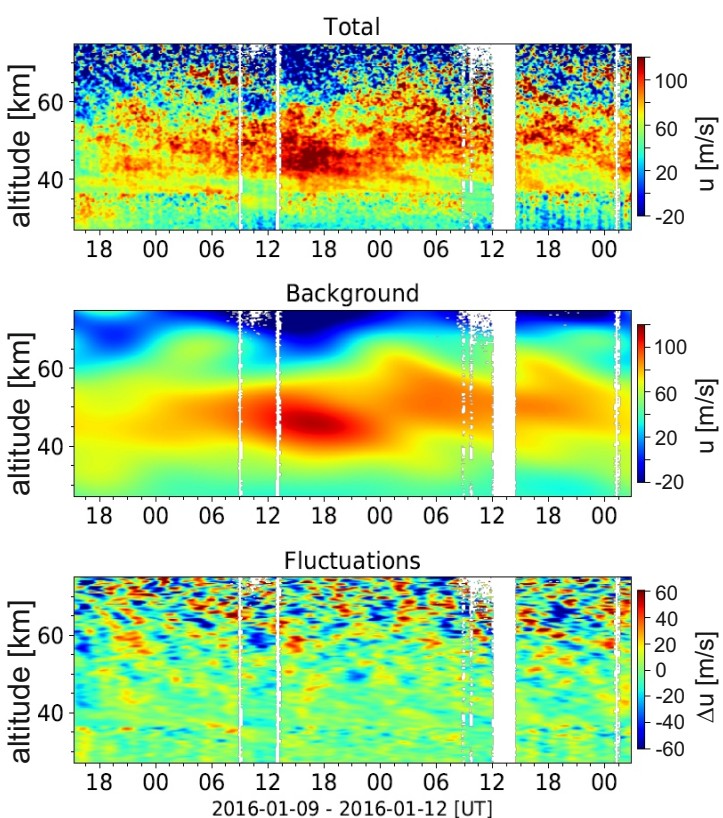

**Figure 3.** The same as Fig. 2 for zonal wind observations.



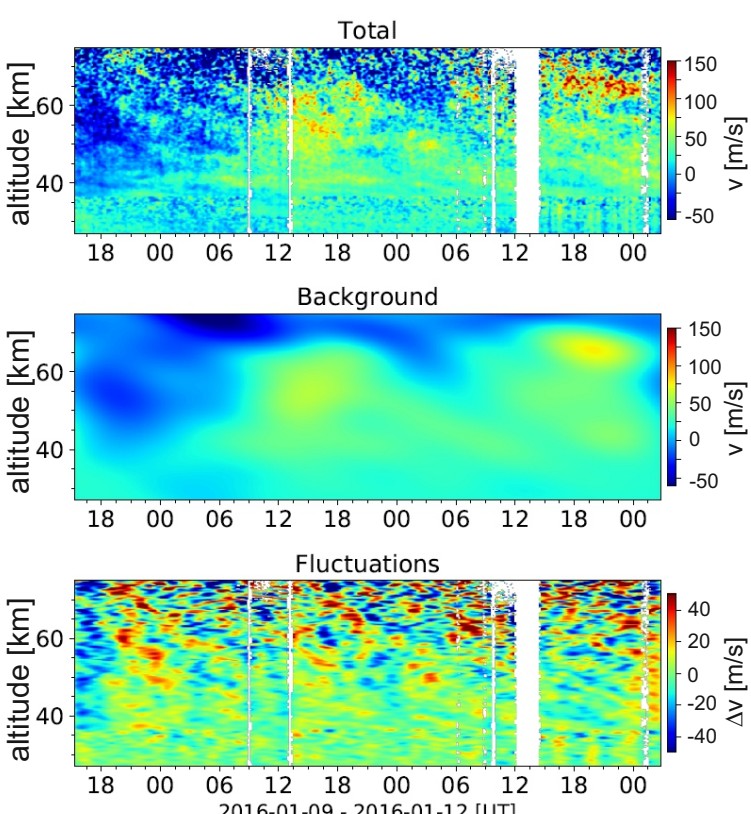

**Figure 4.** The same as Fig. 3 for meridional wind observations.

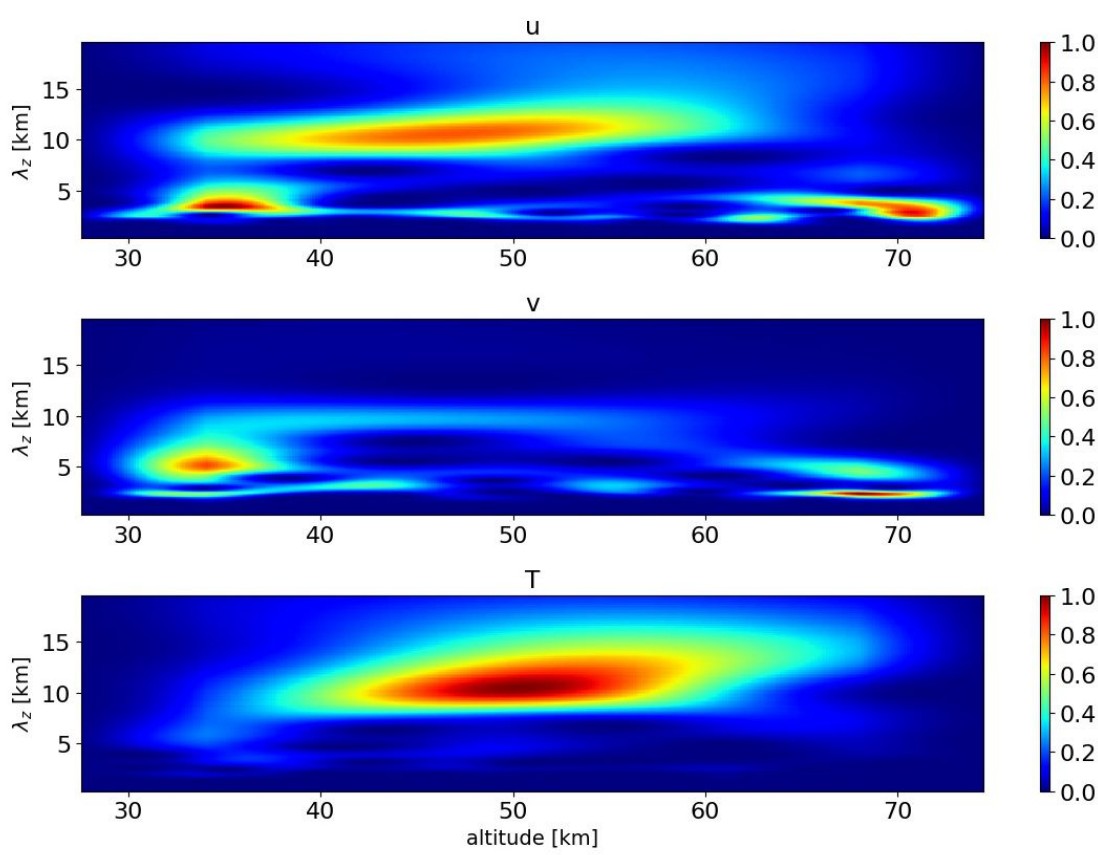

**Figure 5.** Wavelet transform of zonal and meridional wind and temperature fluctuations at 02:07 UT 10 January 2016.





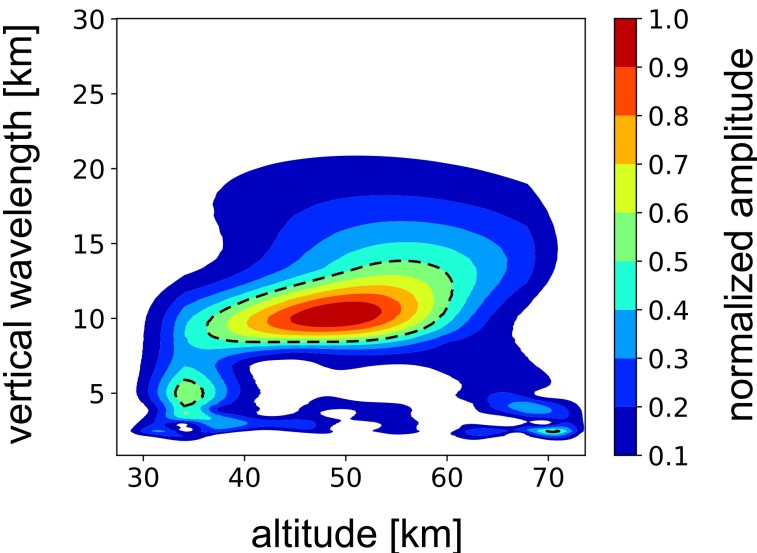

**Figure 6.** Combined wavelet transform of profiles, shown in Fig. 5.

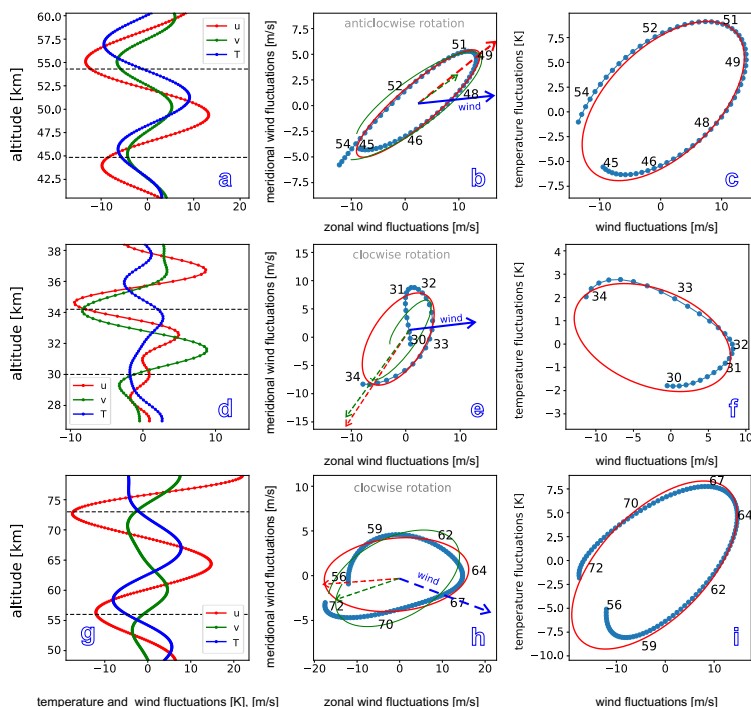

**Figure 7.** (a), (d) and (g) are vertical profiles of observed fluctuations of both wind components and temperature observed at 02:07 UT 10 January 2016, dashed lines mark the altitude range used for the hodographs. Hodographs of: (b), (e) and (h) the zonal wind versus meridional wind fluctuations and (c), (f) and (i) the parallel wind fluctuations ($u'_\parallel$) versus temperature fluctuations. Further quantities of the GW and the background are listed in table 1.





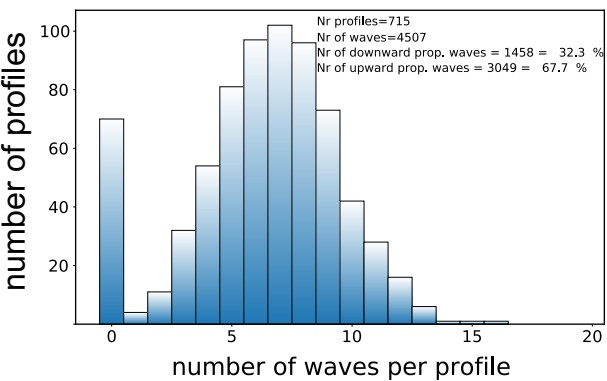

**Figure 8.** Total number of waves obtained per altitude profile for the entire dataset.

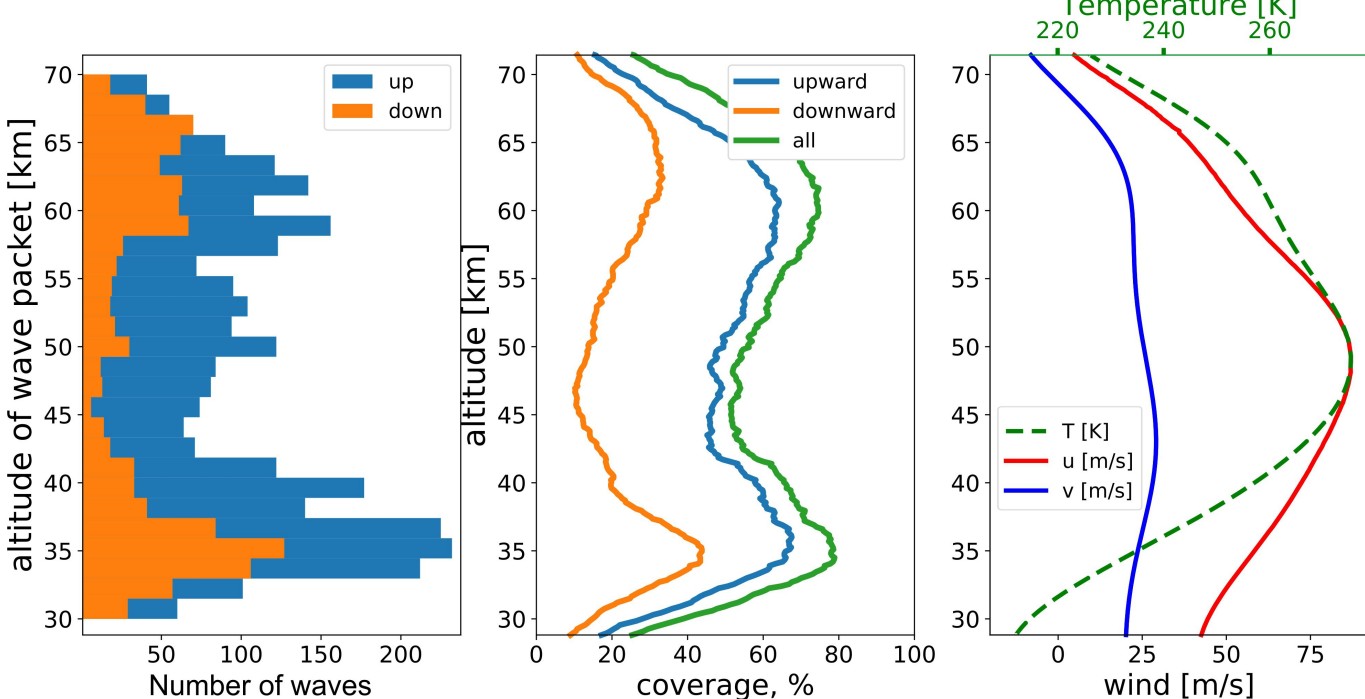

**Figure 9.** Left: Number of waves detected per 1.5 km altitude range bin. Blue (orange) bars mark upward (downward) propagating GW. Middle: Mean coverage by detected waves when taking the altitude extent of the waves into account. The green profile indicates whether any wave was found, whereas blue and orange lines are for up- and downward propagating waves, respectively. Right: Background mean wind and temperature.





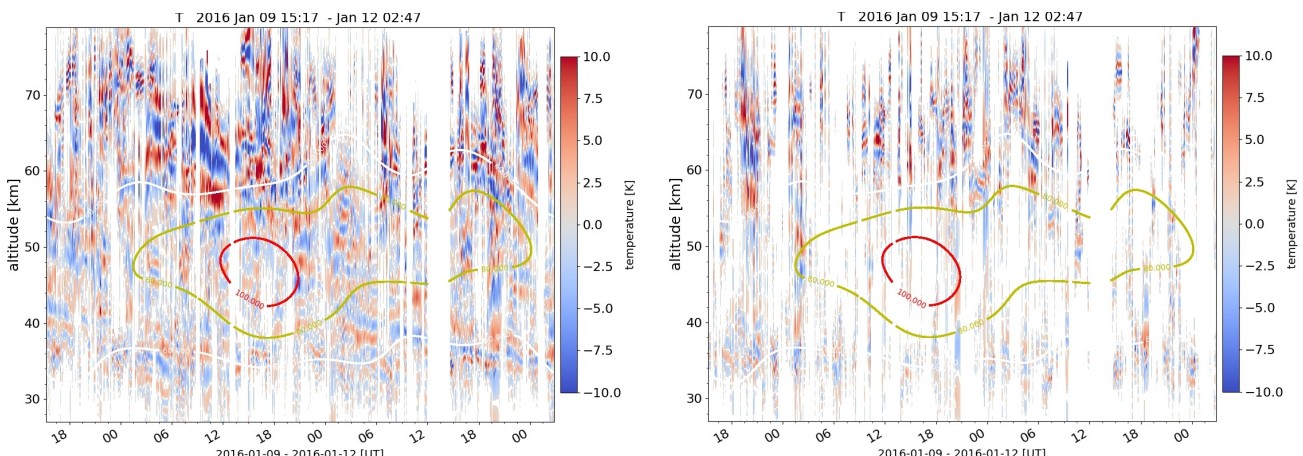

**Figure 10.** Reconstructed temperature fluctuations of upward propagating GW (left pannel) and downward propagating GW (right pannel). Contour lines show the background zonal wind and the numbers on the contour lines are given in m/s.

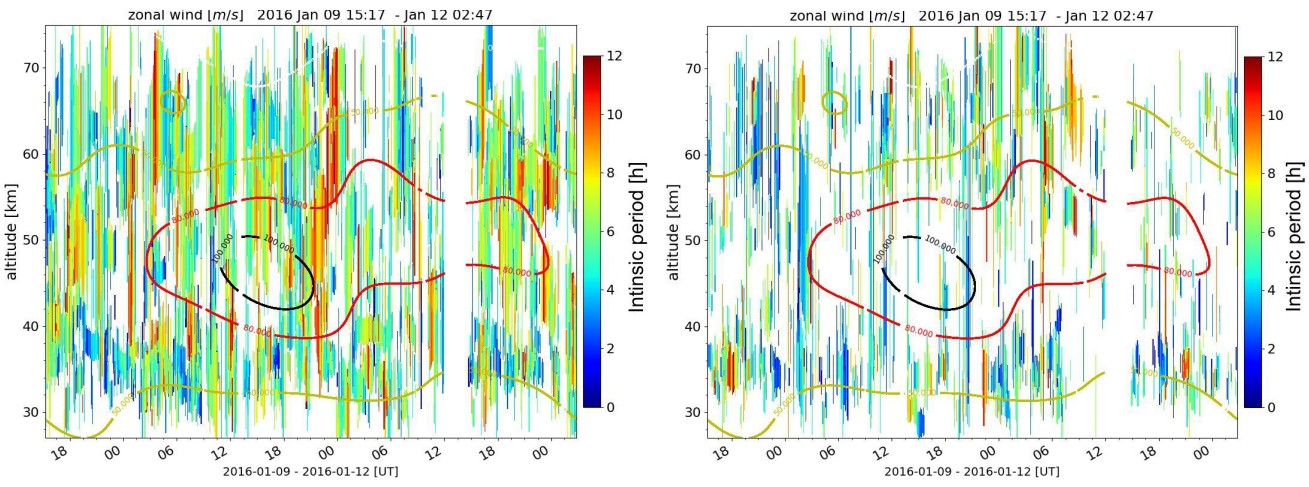

**Figure 11.** Color coded bars show the intrinsic period of upward (left) and downward (right) propagating waves. The length of the bar is given by the extend of the waves. Contour lines show the background total wind, the numbers on the contour lines are given in m/s.





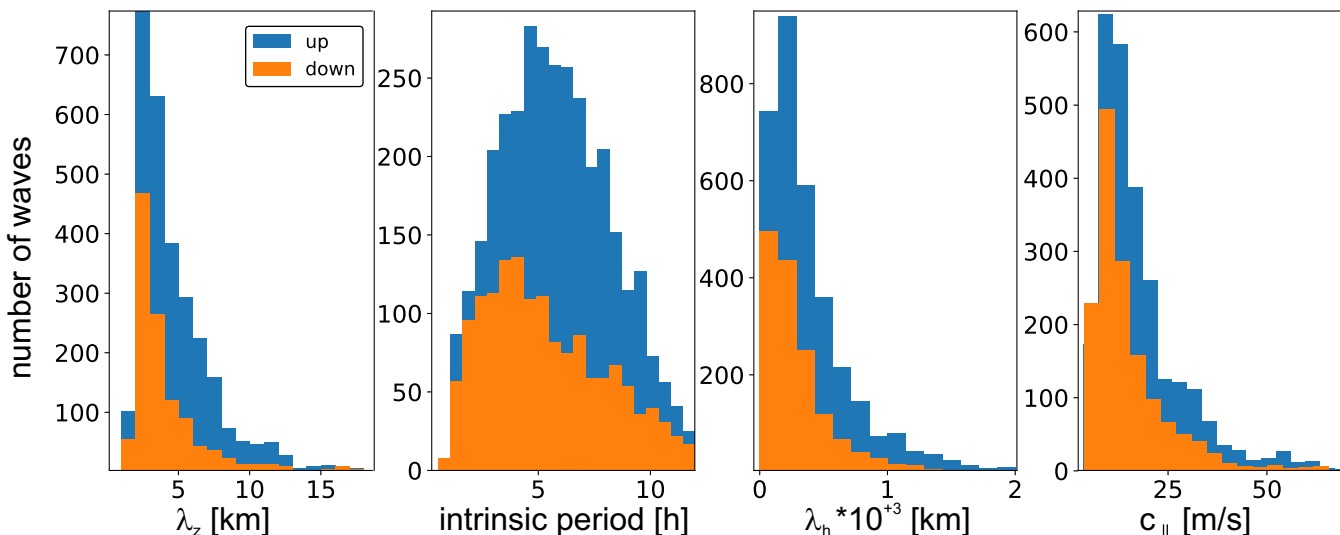

**Figure 12.** Histograms of different GW properties separated for up and downward propagating waves. From left to right: vertical wavelength; intrinsic period; horizontal wavelength; horizontal phase speed. Estimated from equations 1, A5, A9, A10a, respectively.

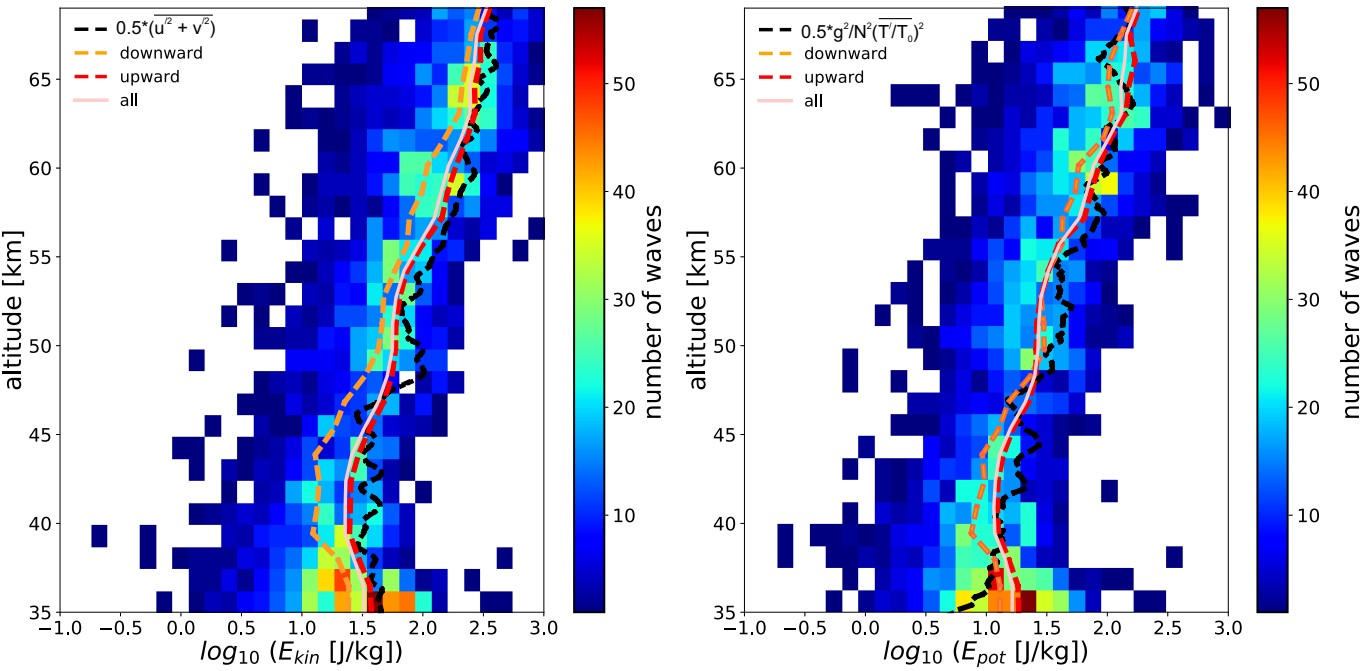

**Figure 13.** Kinetic (left) and potential (right) energies of waves. Colored boxes show 2-D histograms (number of waves per 1.5 km altitude, and 0.15 log(energy) bin). Lines show mean values of whole distribution (pink), upward/downward propagating waves (red/orange dashed) and black dashed lines are energies estimated from the variance of the temperature (left) and wind (right) fluctuations throughout the measurement.





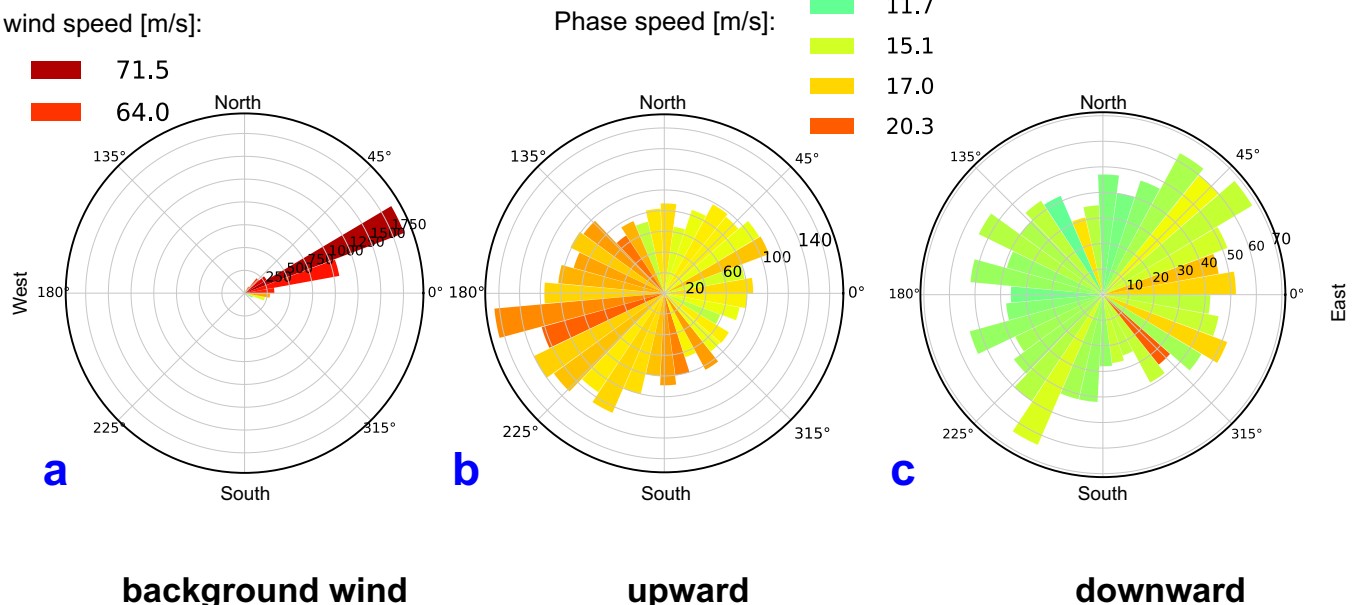

**Figure 14.** Polar histograms of the direction of the background wind (left) and waves for upward (middle) and downward (right) propagating waves. Length of the bars represents the number of waves per $10°$ horizontal direction. The color represents the average wind (left) or average intrinsic phase speeds (middle, right) for the respective directions.

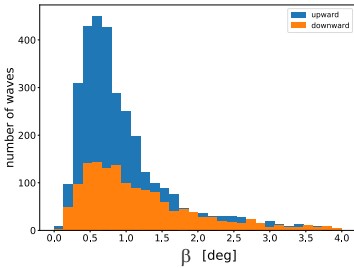

**Figure 15.** Histogram of the (absolute value) of the angle between the group velocity vector and the horizon, separated for up- and downward propagating waves.



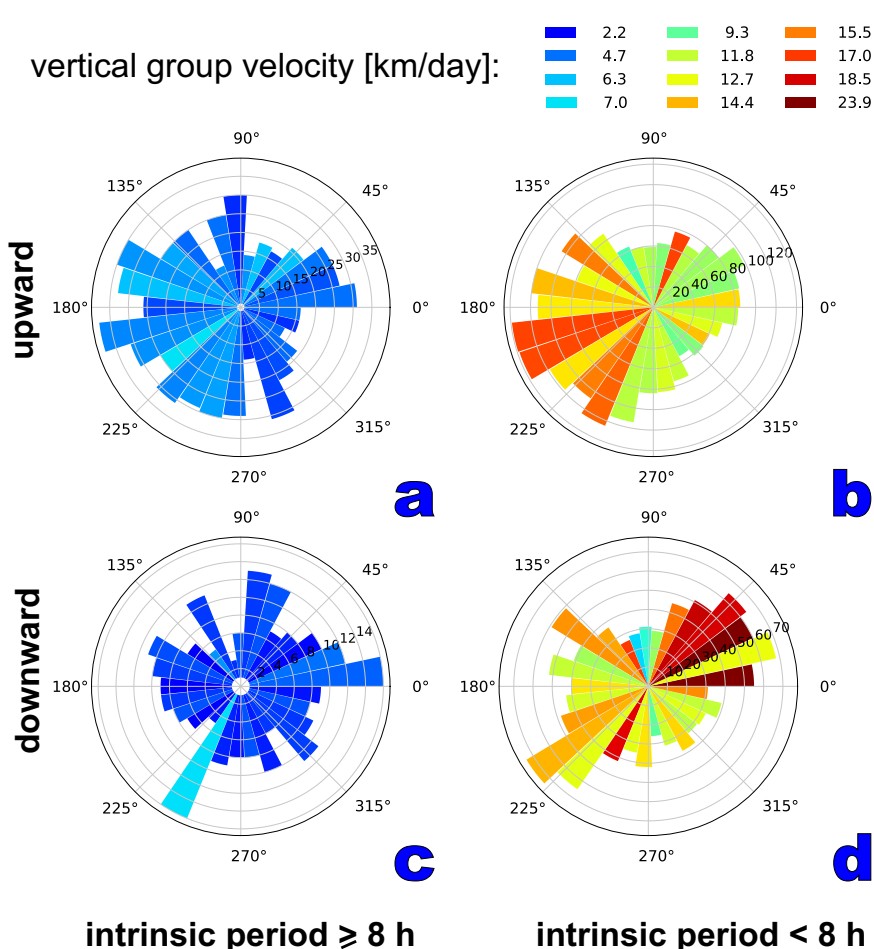

**Figure 16.** Polar histogram of the upward (upper row) and downward (lower row) propagating GW separated for waves with intrinsic periods ≥ 8 hours (left) and <8 hours (right). The length of the bars represents the number of waves per given horizontal direction. The colors represent the vertical group velocity in km/day.



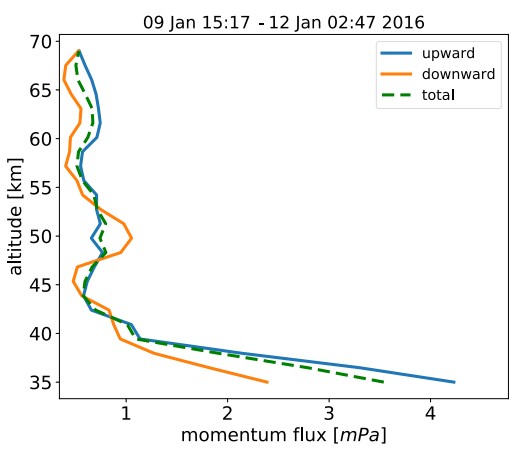

**Figure 17.** Vertical flux of horizontal momentum averaged through all observed hodographs (dashed), upward (blue), and downward (orange) propagating waves.

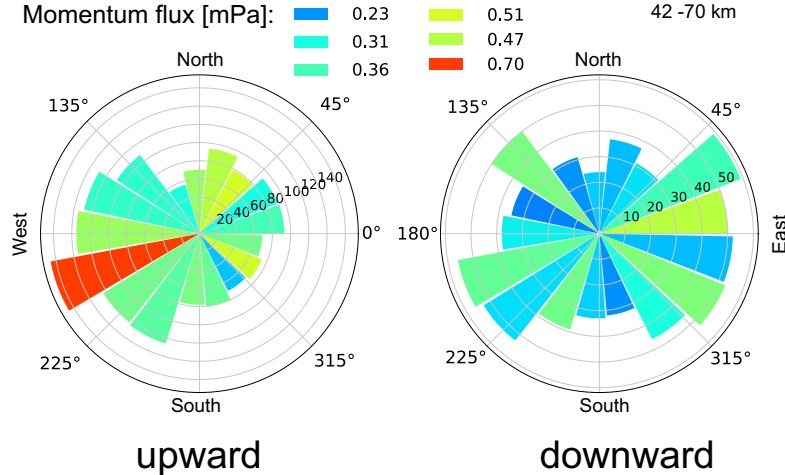

**Figure 18.** Polar histogram of upward (left) and downward (right) propagating waves limited to the altitude range from 42 to 70 km. The length of the bars represents the number of waves per given horizontal direction. Color coded is the average momentum flux in per 20° directional bin in mPa.