# Peer review of "Advanced hodograph-based analysis technique to derive gravity waves parameters from Lidar observations"

_Atmospheric Measurement Techniques, 2019_

## Referee Comment (RC1) · Anonymous Referee #3 · 13 Jul 2019

Advanced hodograph-based analysis technique to derive gravity waves parameters from lidar observations Irina Strelnikova, Gerd Baumgarten, and Franz-Josef Luebken

The paper investigates the use of a hodograph analysis for the detection and characterization of gravity waves in the middle atmosphere. The study employs 30-70 km lidar wind and temperature profiles that were acquired by Rayleigh lidar over 2.5-days of nearly-continuous measurements. The combination of this analysis and this data set provide a unique opportunity to both characterize gravity waves as well as provide a basis for understanding gravity wave retrieval methods and physical interpretations from other measurements. The study should be of interest to a variety of researchers

interested in detection, characterization, and interpretation of atmospheric waves from observations.

The paper is suitable for publication in the discussion forum. However, I have some specific comments that I hope the authors address as they undergo subsequent peer-review.

1) GW Polarization The authors present gravity-wave polarization relations, relating zonal and meridional wind fluctuations (Eqn 2), and temperature and zonal wind perturbations (Eqn 3). In equation 3, the authors claim to use the follow Hu et al., 2002 (Hetal02) and Geller and Gong, 2010 (G&G10). In Hetal02 the authors have a (1/2H) term added to the (I'm) term. In G&G10 the authors suggest that their derivation of a polarization relationship relating relative temperature perturbations to pressure perturbations is based on the assumption that the relative temperature fluctuations are identical to the relative potential temperature fluctuations. Do the authors know how these relationships compare to the formulation based on the ideal gas law that relates relative pressure, density, and temperature perturbations directly? If the authors find insignificant differences, particularly for the inertia-gravity waves in this study, then they could explicitly state that.

2) Intrinsic and Observed Frequencies The authors have complete wind measurements that allows them determine both the observed and intrinsic frequencies of the waves. Can the authors add the observed frequency of the waves to the list of results for the three waves in Table 1. In general, can the authors comment on the relationship between the observed and intrinsic frequencies for the waves they have characterized.

3) Identifying Waves The study reports 4507 quasi-monochromatic waves. However, if I understand it right the study has found 4507 snapshots of some number of waves based on hodograph analysis of ∼240 profiles. The authors discuss that the individual waves persist in their presentation of temperature fluctuations and intrinsic periods in Figures 10 and 11. Can the author quantify the life-time of the waves in the data

set? There have been discussions in the literature about how many gravity waves are present and the intermittency of gravity waves. This study has the opportunity to address the life-time of gravity waves, particularly relating it to the spatial and temporal scales of the waves, that would address a variety of questions about wave dynamics and evolution.

---

## Referee Comment (RC2) · Anonymous Referee #4 · 25 Jul 2019

This is an interesting study introducing a new evaluation method for data which continuously record both winds and temperatures - currently the case only for very specialized lidar systems. The method is demonstrated for a period recorded 9-12 January by the IAP ALOMAR lidar system. The results are novel and merrit publication in AMT. However, GW momentum flux is the more interesting and conclusive quantity and some more information with regard to the altitude development should be given, see comments below.

Major comments:

You produce most of your diagrams for "number of waves". However, from the dynam-

ics point of view GW pseudomomentum flux is most relevant. It would be very helpful if you add a second row to Figure 12 where you plot the total absolute momentum flux of the waves in a wavelength bin. (You could normalize that in a way that the total GWMF of all waves (up + down) is normalized to 1. and keep that same normalization also for up and down separately). Same for F14 and F15.

The vertical wavelengths you observe are rather small. Starting from the very first work on saturated spectra (Smith, Fritts VanZandt, 1987) we have indication that the wavelength of the maximum in the distribution shifts to longer wavelengths at higher altitudes. Follow-up work by e.g. Gardener et al. and the general concept of the Warner & McIntyre scheme infer a power law for this. You can put in several observations by e.g. radio sondes, rockets ... to calibrate this. Then you would expect something like 2km in the lower stratosphere, 10-15km in the mesopause region and accordingly ∼5km around the stratopause. The satellite data certainly have a long-bias, but they confirm the increase of typical wavelengths with altitude. Compared to this you have 2km which one would expect for the low stratosphere in a data set which goes up to the mesopause. One reason may be that you give your histograms for number of waves only. Still it would be good to see some vertical profile of avereage vertical wavelengths, normal average as well as GWMF weighted, up + down searately, so for profiles in total.

Phase speed is approx proportional to vertical wavelength. The observational filter for airglow is totally different (lz>10km), so no wonder that phase speeds are much higher. There is a wealth of literature on phase speeds from different sources (convection, spontaneous imbalance, ...). Maybe it is more worthwhile to compare to that. The phase speed diagram kind of seems to exclude convection as dominant source here. Still there is the general issue about the short vertical wavelengths.

Minor comments and technical suggestions:

P2L1 Suggest to omit colloquial phrase: The problem is that

P2L4 these models need to rely on various parametrizations.

P2L18 information about GWs (), but they base solely on temperature observations.

P2L20 Did the Shigaraki radar not provide some winds? If so: provide high-resolution wind

P2L27 legacy -> established ?

P2L31 Would be nice, if we were sure about this. In our analysis technique we aim solely at such fluctuations which are generated by GW.

P2L35 aimed -> aim

P3L20 and afterwards smoothed

P4L8 Please check and reformulate the sentence

P4L26 For an hodograph you need a certain altitude range, so it is not 'in the center'. As you specify the altitude range below: 'around the'

P4L30 several oscillation periods

P5L3 remove

P5L6 Coriolis

P6L2 procedure plays a key ... and may even lead ...

P6L12 which supposedly are produced

P6L13 Please describe all panels: The upper panels show the measured data, the middle ...

P6L16 made -> performed

P6L23 as the one most

P6L25 dissipation ; refraction may also change the amplitude, conserving wave action

flux Under the assumptions of conservative propagation and a constant background,

P6L29 Sorry, I don't understand this: You cannot analyze small-scale structures at low altitudes in the atmosphere? Please explain or reformulate.

P7L17 So you don't do that? Why not?

P7L20 or 3) the wave source process generates waves with a frequency changing in time Just a remark, no request to discuss this

P7L26 Which step 4.2 are you talking about? There is nothing here to indicate which steps form the whole algorithm

P8L2 which cover an altitude range of approx. 50km and thus much longer than a wavelength and the expected scale of amplitude variations.

P13L9 Please include a sentence that when the average energy is about the same but you have less downward waves also the total energy of downward waves is less.

F13 I like that figure, but it would be great if you could add two more panels: Vertical wavelengths and GWMF.

P14L1 And this is really puzzling! You have most of the waves and the momentum flux propagating against the wind and the wind velocity increases at higher altitudes, so no critical level filtering. Vertical wavelengths then should increase which leads to lower amplitudes at same GWMF, so no saturation expected either. Reflection? It would be good to know at least which parameter changes most (wavelengths, amplitudes ...) as to produce this result. Or do you have an edge effect in your retrieval or your method?

P14L16 Corwin Wright does something similar for his AIRS analyses

P15L8 I do not think that this is true! It would needed to be shown, anyway. There should be places where temperature and wind amplitudes overlap and I would expect phase shifts of $\sim 90°$ between winds and temperatures. Just watch one of the tidal movies. Here, you have likely removed the tides by the 15km vertical cut-off, though.

---

## Referee Comment (RC3) · Anonymous Referee #1 · 25 Jul 2019

This paper presents an advanced analysis method for lidar wind and temperature data to derive gravity wave properties. The analysis method consists of 2D-FFT to define the background, a scaling of the amplitudes to remove the influence of growing amplitude with altitude/decreasing density, wavelet analysis in combination with a cosine fit to isolate monochromatic waves, and subsequent hodograph analysis to determine wave properties. The approach is demonstrated for a case study over Northern Norway.

The paper addresses the issue of deriving gravity wave properties from ground-based lidar measurements of wind and temperature in the middle atmosphere. Deriving gravity wave properties is an issue which affects all sorts of measurements from ground-

based to spaceborne. Several analysis methods have been developed in the past. They all have their limitations and assumptions which affect the derived gravity waves properties. Development and adaption of gravity wave analysis methods for present types of measurements is an important task. Detailed descriptions of state of the art methods are required for reproducibility and comparability. Therefore, the paper is of interest to the gravity wave community. The presented case study nicely shows what kind of statistics the analysis can reveal with the advantage that it could be applied to even longer time series of lidar data. However, there are some issues the authors should address before final publication.

Major comments:

1) The current structure of the paper makes it hard to follow the story at some point. This is especially true for section 3 (theoretical introduction) which already presents one part of the analysis method, i.e. hodograph analysis. I suggest to include the content of current section 3 in the next section and place the details about hodograph analysis in the respective subsection.

2) Scaling seems to be an essential step of the analysis (section 4.2). Here, you can refer to Wright et al. 2017 who applied the scaling to satellite data. They used a reference altitude in the middle of their observations (41 km). Can you tell if the scaling altitude has an influence on your results? One may question if it's reasonable to scale amplitudes to surface values ($z=0$) for measurements starting above 25 km.

Wright, C. J., Hindley, N. P., Hoffmann, L., Alexander, M. J., and Mitchell, N. J.: Exploring gravity wave characteristics in 3-D using a novel S-transform technique: AIRS/Aqua measurements over the Southern Andes and Drake Passage, Atmos. Chem. Phys., 17, 8553-8575, https://doi.org/10.5194/acp-17-8553-2017, 2017.

3) I don't fully understand how the fitting process of the cosine functions works. Please, try to clarify. What is prescribed in the first guess? Where do the values come from? See comments P8, L3; P8, L27; P9, L15

Minor/detailed comments:

P1, L12-15: It doesn't seem necessary to give/repeat details about the (hodograph) technique here. . . .We identified 4507 quasi monochromatic waves. In the vicinity of the polar night jet. . .

P2, L2: define small scale (horizontally, vertically)

P2, L7: high resolution numerical modelling is also a useful tool; please remove "only"

P2, L24: again, what is meant by "these small scale waves"

P3, L4: I don't think that geophysical meaningful results are enough to justify the capability of the method at this point. I suggest to simply go with "Finally in section 5, the capability of the method is demonstrated with continuous ALOMAR lidar data during a four day period in 2016.

P5, L3: comment shows up: "%begin equation"

P6, L7: This means your algorithm doesn't take into account stationary waves because they are assigned to the background. Should be mentioned here.

P6, L13: ". . .which might only be produced by gravity waves. . .": I don't think this is true and you already mentioned in your introduction that wave structures must be distinguished from e.g., turbulence. The fluctuations need to follow the GW-dispersion relation which is hard to prove in measurements as one usually lacks either vertical or horizontal information of the wave structure.

P6, L19: ". . .skip this step from the analysis.": this conclusion doesn't make sense to me. Don't you need fluctuations of u, v, T for all the analysis?

P7, Fig. 5: Did you apply zero-padding to the data? You should explain and include the cone of influence of the wavelet analysis. It limits the interpretation of signatures at the edges and the true vertical extent of packages with longer wavelength.

P8, L3: I cannot fully follow the description in this paragraph. You start with a first guess from the scalogram for z0 and vertical wavelength (are you automatically searching for the maxima?). Your fitting reveals intrinsic frequency and propagation direction + corrected z0 and vert. wavelength? The equations for T', u', v' depend not only on z0 and vertical wavelength. Which values are you using for the wave packet width and intrinsic frequency? Propagation direction is calculated afterwards using Eq. A4. Isn't the conclusion then that Eq. A4 is not well performing for intrinsic periods larger than 1h OR the whole fitting process including z0 and vertical wavelength brings some uncertainty?

P8, L12: "For low frequency GW, i.e. those with periods close to the Coriolis period ($2\pi/f$) the fluctuations reveal a circle." This does not agree with the fact that hodograph method/stokes analysis is especially used and appropriate for gravity waves with intrinsic frequencies close to the inertial frequency (<10f), i.e. showing an ellipse in the hodograph?

P8, L27: "Additionally we calculate a vertical wavelength by requiring the hodograph to close the full 360° cycle." How is this done? Why all the effort to correct the vertical wavelength in the previous step if you could use this value anyway?

P8, L1: Did you account for the influence of transverse-shear on the axial ratio of the ellipse? *correction given in: Vincent R.A., Allen S.J., Eckermann S.D. (1997) Gravity-Wave Parameters in the Lower Stratosphere. In: Hamilton K. (eds) Gravity Wave Processes. NATO ASI Series (Series I: Environmental Change), vol 50. Springer, Berlin, Heidelberg *effect was pointed out by Hines, C. 0., 1989: Tropopausal mountain waves over Arecibo: A case study, J. Atmos. Sci., 46, 476-488

P9, L15: "That is, the dominating frequency is used as a zero guess for the fitting of Eqs. 1 to derive exact values of z0 and $\lambda z$.." Now, I am totally confused (see comment P8, L3)

P11, L1: I think you already have demonstrated how the method works with real data

profiles. This section now shows "Finally, this algorithm for a single point in time is subsequently applied to all time points of the entire data set shown in Fig. 2, 3 and 4." as you say at the end of the previous section. Maybe you can just shift this sentence to the beginning of this section.

P12, L1-8: I recommend to put this paragraph prior to the up/downward discussion of literature.

P12, L6: Any physical explanation for this finding? Enhanced vertical wavelength due to high wind speed?

P12, L10: No scaling of the amplitudes? To enhance the visibility at lower altitudes compared to higher altitudes, it may be useful scale amplitudes in Fig. 10.

P14, L14-15: But didn't you mention earlier that the sensitivity of your analysis to the chosen background is small?

P14, L17: compare comment P6, L19

P14, L21: Wright et al 2017, see major comment 2

P15, L12: "additional robust algorithm to pick out wave packets automatically". Isn't this what your algorithm does already as implicated by "our algorithm resolves many more GWs than it can be inferred by manually applied hodograph technique"? Please clarify.

P15, L14: holographs should be hodographs

---

## Referee Comment (RC4) · Anonymous Referee #2 · 29 Jul 2019

The author presents an algorithm that simultaneously conducts 2D-FFT and hodograph technologies on single time lidar profiles to derive GW parameters. The technique is almost completely based on the spatial structure in vertical direction of these gravity waves, and it is amazing to see the so many number of gravity waves (4507) existed during 60 hours of lidar operation. To test this approach, I suggest the author combine the reconstructed upward and downward perturbations for temperature and wind fields in Figure 10 and 11, and compared with the real perturbations shown in Figure 2,3,4. The total reconstructed perturbations should be quite close to the measured perturbation. It would also be helpful, if the author could test the results in temporal domain by looking at the wavelet (or lombscargle) results (in time) of the real perturbation and

the total reconstructed perturbation, to see if the algorithm does not lose the temporal variations of these waves. I would also like to know the measurement uncertainties during this lidar campaign, although I am aware that the author treats (weights) every lidar measurement the same (without error?). This is a numerical technique based upon lidar observations, so, I think it is important to know the data quality.

———————————————

---

## Short Comment (SC1) · 19 Aug 2019

This review looks identical to the previous one (before discussion phase). Is it so on purpose? Or some technical mistake happened? Thank you for clarifying.

---

## Author Comment (AC3) · 9 Oct 2019

The comment was uploaded in the form of a supplement:
https://www.atmos-meas-tech-discuss.net/amt-2019-79/amt-2019-79-AC3-supplement.pdf

---

## Author Comment (AC4) · 9 Oct 2019

The comment was uploaded in the form of a supplement:
https://www.atmos-meas-tech-discuss.net/amt-2019-79/amt-2019-79-AC4-supplement.pdf

---

## Author Response (AR1)

**Response to reviewers' reports on the paper amt-2019-79 Advanced hodograph-based analysis technique to derive gravity waves parameters from Lidar observations**

Irina Strelnikova[1], Gerd Baumgarten[1], and Franz-Josef Lübken[1]

[1]Leibniz-Institute of Atmospheric Physics at the Rostock University, Kühlungsborn, Germany

*Correspondence to:* Irina Strelnikova (strelnikova@iap-kborn.de)

We appreciate the reviewers' constructive comments and their positive judgment on our paper. We have taken the reviewers' suggestions into account when preparing the revised version of our manuscript.

However, we would like to make a general comment. This paper is submitted to AMT with purpose to describe a method of analysis. We demonstrate on a data set how this method works. We also demonstrate how to obtain extended set of GW
5   parameters and summarize equations and assumptions used for estimation of different parameters. We do not claim that this data set represents a "typical" situation in polar winter season. Thus, in this manuscript we try to avoid making general conclusions like behavior of momentum flux or vertical wavelength as a function of altitude or any other parameter. We are currently working on another manuscript where a larger data set is analyzed by this method. We will take into account the corresponding suggestions of referees when preparing the next manuscript.

10   In the following we address the comments of all reviewers point by point.

**To Referee 1**

**1) The current structure of the paper makes it hard to follow the story at some point. This is especially true for section 3 (theoretical introduction) which already presents one part of the analysis method, i.e. hodograph analysis. I suggest to include the content of current section 3 in the next section and place the details about hodograph analysis in the**
15   **respective subsection.**

To address this reviewer's comment we revised the sections 3 and 4 to make the theory and the analysis technique to be clearly separated.

**2) Scaling seems to be an essential step of the analysis (section 4.2). Here, you can refer to Wright et al. 2017 who applied the scaling to satellite data. They used a reference altitude in the middle of their observations (41 km). Can you**
20   **tell if the scaling altitude has an influence on your results? One may question if it's reasonable to scale amplitudes to surface values (z=0) for measurements starting above 25 km.**

To address this reviewer's comment we added two notes, in Sec. 4.2 and 4.7, respectively.

In Sec. 4.2 **Scaling of fluctuations**:

*Note however, if further analysis requires treatment of fluctuation amplitudes, this scaling must be either taken somehow into account (e.g., by appropriate normalization) or removed (by applying inverse scaling) as we do in Sec. 4.7.*

In Sec. 4.7 **Calculation of GW parameters**:

*Note, that as mentioned in Sec. 4.2, at this point the fluctuation amplitudes must be rescaled back to their original growth rate with altitude using the derived scaling parameter $\varsigma$, to legitimate their use for e.g., estimation of wave energy.*

Here we give a more detailed explanation which, we believe would disimprove the readability of our manuscript.

The scaling altitude does not affect the final results neither in the analysis used by Wright et al. (2017) nor in the analysis shown in our manuscript. The reason for that is the inverse rescaling applied to the fluctuations before their actual use:

Wright et al. (2017): *"This restores the true height-scaling of the measured wave amplitudes, typically exponentially increasing with height"*.

Note, however, that our scaling approach is different to what has been used by Wright et al. (2017). Namely, instead of using $\exp((z - z_0)/(2H))$, we apply scaling $\exp(z/(\varsigma H))$, where parameter $\varsigma$ is individually (and automatically) adjusted to every profile at step 4.2 (scaling) and is further used for inverse scaling at step 4.7 (calculation of GW parameters).

In the approach used by Wright et al. (2017), the choice of $z_0$ can only influence amplitude of fluctuations if it increases with altitude not as $\exp((z - z_0)/(2H))$, but $\exp(z/(\varsigma H))$. Figs. 1, 2, 3 demonstrate the simulated Wright et al. (2017)'s scaling process.

[Figure]

**Figure 1.** Vertical profile of temperature fluctuations. Black (blue) dashed line was estimated for $z_0 = 41 km$ ($z_0 = 20\ km$)

[Figure]

**Figure 2.** The same as in Fig. 1, but in logarithmic scale. Orange line demonstrated wave amplitude used in simulations.

[Figure]

**Figure 3.** Vertical profile of temperature fluctuations normalized by $\exp((z - z_0)/(2H))$, where $z_0 = 41 km$ ($z_0 = 20\ km$) for black (blue) lines

Here we simulated GW with increase of its amplitude with height as $1.0/\sqrt{density}$, where $density$ was taken from the NRL-MSISE00 model. The continuous blue line shows this GW in Fig. 1 and 2 in linear and log scale, respectively. The GW-amplitude increase of $1.0/\sqrt{density}$ as it is derived from the MSIS data is shown by the orange line in Fig 2. Note, that we used MSIS density profile for January because it reveals more pronounced difference between $\exp((z - z_0)/(2H))$ if compared to summer. The dashed blue and dashed black lines in Fig. 1 and 2 show the scaling factor $\exp((z - z_0)/(2H))$ derived for $z_0 = 20$ and $z_0 = 41$ km, respectively. Whereas orange line represents the natural GW-amplitude increase. As it

is seen in logarithmic scale (Fig. 2) the both dashed lines (i.e., for $z_0 = 20$ and $z_0 = 41$ km) are parallel to each other and they differ only because the increase of the "natural" (=MSIS in this case) GW-amplitude varies with altitude. This variation produces such altitude-dependent difference between $\exp((z - z_0)/(2H))$ for different $z_0$. In summer case these both lines will be identical and no difference for different $z_0$ will be observed.

5      After applying these two different scalings (derived for different $z_0$) we get fluctuations shown in Fig. 3 as dashed blue and dashed black lines for $z_0 = 20$ and $z_0 = 41$ km, respectively. These two profiles of scaled fluctuations reveal similar behavior as far as altitude dependence is concerned (Fig. 3).

**One may question if it's reasonable to scale amplitudes to surface values (z=0) for measurements starting above 25 km.**

10      This question arise more likely because we use a function $\exp(z/(\varsigma H))$ for normalization (i.e. $z_0 = 0$). We can rewrite $\exp((z - z_0)/(\varsigma H))$ normalization as $\exp((z)/(\varsigma H)) \cdot \exp((-z_0)/(\varsigma H))$. Since we assume, that $\varsigma$ and $H$ are constant at given altitude range, we can rewrite this normalization as $const \cdot \exp(z/(\varsigma H))$. Thus, we can divide our observations by this $const$ and later multiply results by the same $const$. Finally, results will be the same.

     **3)I don't fully understand how the fitting process of the cosine functions works. Please, try to clarify. What is pre-**
15      **scribed in the first guess? Where do the values come from? See comments P8, L3; P8, L27; P9, L15**

     To address this reviewer's comment we completely rewrote the Sec. 4.4 (Fitting of linear wave theory) to make the description of fitting process better understandable (see revised version of the manuscript).

     **Minor/detailed comments:**

     **P1, L12-15: It doesn't seem necessary to give/repeat details about the (hodograph) technique here. ...We identified**
20      **4507 quasi monochromatic waves. In the vicinity of the polar night jet...**

     Changed as suggested.

     **P2, L2: define small scale (horizontally, vertically)**

     Improved as suggested:

*... waves with horizontal wavelengths typically shorter than 1000 km.*

25      **P2, L7: high resolution numerical modelling is also a useful tool; please remove "only"**

     Changed as suggested.

     **P2, L24: again, what is meant by "these small scale waves" P3, L4: I don't think that geophysical meaningful results are enough to justify the capability of the method at this point. I suggest to simply go with "Finally in section 5, the capability of the method is demonstrated with continuous ALOMAR lidar data during a four day period in 2016.**

30      To address this reviewer's comment we made it more specific in the text:

*...lidar technology give us new possibilities to study GW experimentally on a more or less regular basis and resolve spatial sales of 150 m in vertical and temporal scales of 5 min*

     **P5, L3: comment shows up: "% begin equation"**

     corrected as suggested

**P6, L7: This means your algorithm doesn't take into account stationary waves because they are assigned to the background. Should be mentioned here.**

The sentence that confused reviewer was: "*We define the background as wind or temperature fluctuations with periods longer than 12 hours and vertical wavelengths longer than 15 km.*"

Thus, our algorithm indeed excludes stationary waves if they have vertical wavelength longer than 15 km. Stationary waves with shorter vertical wavelength are not removed. This can be seen in Figures 2-4. Middle panels demonstrate obtained backgrounds and lower panels demonstrate remained fluctuations. Structures defined as background do not reveal something like "stationary waves". On the contrary, remaining fluctuations demonstrate in some places such a behavior. For example, some fluctuations below ∼40 km look like stationary waves (especially in meridional wind, lower panel of Fig. 4) To note again, this is the advantage of the 2D-FFT method if applied for the background removal. See also our previous response to reviewers, where we had a detailed discussion about background definition (Appendix B), where different approaches were discussed.

Note also, that we changed the colormap when preparing the new version of manuscript in order to demonstrate fluctuations in more appropriate way.

**P6, L13: "...which might only be produced by gravity waves...": I don't think this is true and you already mentioned in your introduction that wave structures must be distinguished from e.g., turbulence. The fluctuations need to follow the GW-dispersion relation which is hard to prove in measurements as one usually lacks either vertical or horizontal information of the wave structure.**

To address this reviewer's comment we rephrased the sentence to make it clear that: *After subtracting the derived background from the original measurements we obtain the wind and temperature fluctuations which have periods shorter than 12 hours or wavelengths sorter than 15 km.*

**P6, L19: "...skip this step from the analysis.": this conclusion doesn't make sense to me. Don't you need fluctuations of u, v, T for all the analysis?**

To avoid such a confusion we rephrased this sentence as follows: *The new technique is not sensitive to the background derivation schemes and may use simpler background calculations like constant values in time.*

**P7, Fig. 5: Did you apply zero-padding to the data? You should explain and include the cone of influence of the wavelet analysis. It limits the interpretation of signatures at the edges and the true vertical extent of packages with longer wavelength.**

We agree with the reviewer that the wavelet transform would reveal limitations connected to the finite length of data set like edge effects etc. We believe that after significant revision of Sec. 4 (as was requested by the reviewer in his/her major comments above) it should be clear now from the text, that the wavelet analysis is only used for estimating initial guess for the further and more robust part of the analysis. That is, the next after wavelet steps do refine the picture and yield more details than can be inferred from the wavelet analysis.

**P8,L3: I can not fully follow the description in this paragraph. You start with a first guess from the scalogram for z0 and vertical wavelength (are you automatically searching for the maxima?). Your fitting reveals intrinsic frequency and propagation direction + corrected z0 and vert. wavelength? The equations for T', u', v' depend not only on z0**

**and vertical wavelength. Which values are you using for the wave packet width and intrinsic frequency? Propagation direction is calculated afterwards using Eq. A4. Isn't the conclusion then that Eq. A4 is not well performing for intrinsic periods larger than 1h OR the whole fitting process including z0 and vertical wavelength brings some uncertainty?**

To address this reviewer's comment together with major comment above we rewrote the section 4.4. In particular, we made it clear in the text, that the uncertainty connected to direct application of Eq. A4 to noisy data can be avoided if we apply next steps in our algorithm instead. Namely, we propose to apply the hodograph method to the extracted wave packets to precisely derive further wave parameters.

**P8, L12: "For low frequency GW, i.e. those with periods close to the Coriolis period $(2\pi/f)$ the fluctuations reveal a circle." This does not agree with the fact that hodograph method/stokes analysis is especially used and appropriate for gravity waves with intrinsic frequencies close to the inertial frequency (<10f), i.e. showing an ellipse in the hodograph?**

If coriolis parameter is equal to intrinsic frequency, i.e. $f = \widehat{\omega}$ in Eq. 2, we get $\widehat{v}_\perp = -i\widehat{u}_\parallel$. Alternatively, if we use Eq. 7.68 from Holton (2004) we can rewrite wave fluctuations as:

$$u' = |\widehat{u}| \cdot cos(kx + mz - \widehat{\omega}t) \tag{1a}$$
$$v' = |\widehat{u}| \cdot sin(kx + mz - \widehat{\omega}t) \tag{1b}$$

That is, equal amplitudes and phase shift of $\pi/2$ means circle. At ALOMAR location $2\pi/f \simeq 12.8$ h. This means, that GW with intrinsic periods close to $\sim 12$ h reveal rather circle than ellipse.

**P8, L27: "Additionally we calculate a vertical wavelength by requiring the hodograph to close the full $360\circ$ cycle." How is this done? Why all the effort to correct the vertical wavelength in the previous step if you could use this value anyway?**

To address this reviewer's comment we improved the description of this procedure: *This correction to the vertical wavelength is found by forcing the hodograph to close the full 360 ° cycle and calculating the additional vertical length resulted from this extra rotation.*

We also note here, that the hodograph technique is very sensitive to the data quality and, in particular, to such specific difficulties like limited dataset or insufficient resolution. That is why we developed this more extensive algorithm of GW-analysis that combines different techniques and uses their advantages when we believe it is more appropriate. Hodograph alone usually fails if its rotation is considerably smaller than 360 °. Also, by iterating wave fitting and hodograph we make another consistency check which improves the robustness of our analysis.

**P8, L1: Did you account for the influence of transverse-shear on the axial ratio of the ellipse? *correction given in: Vincent R.A., Allen S.J., Eckermann S.D. (1997) Gravity-Wave Parameters in the Lower Stratosphere. In: Hamilton K. (eds) Gravity Wave Processes.**

Vincent et al. (1997) concluded, that this effect is not significant ($\sim 6\%$ in winter). From our data analysis we derived similar conclusion. Moreover, since background wind during our observations was restricted to range of azimuth from $\sim 0°$ to $\sim 45°$, transverse shear was restricted to ranges of azimuth from $\sim 90°$ to $\sim 135°$ and from $\sim 270°$ to $\sim 315°$, where amount

of observed waves is minimal. Thus, the quantitative effect of such correction is not significant and only small fraction of detected waves is affected by this correction. For test purposes, we applied such corrections, but in plots, shown in the current manuscript it would be hard to see any differences. On the other hand, Hines (1989) introduced such correction for wind profiles without background removal. Since we remove variable background wind, some effect from vertical displacement due to background wind gradient can be attributed to background and hence, has no influence on the observed ratios. Thus, in order to apply this correction to our data, we have to demonstrate, that correction introduced by Hines (1989) is meaningful for our data analysis. Since impact of such correction (if applied) is negligible, we decided to not include it in our algorithm (i.e., our results are without correction).

**P9, L15: "That is, the dominating frequency is used as a zero guess for the fitting of Eqs. 1 to derive exact values of $z0$ and $\lambda_z$.." Now, I am totally confused (see comment P8, L3)**

We are grateful to reviewer for the careful reading. This is indeed a typographic error and it is now corrected (removed).

**P11, L1: I think you already have demonstrated how the method works with real data profiles. This section now shows "Finally, this algorithm for a single point in time is subsequently applied to all time points of the entire data set shown in Fig. 2, 3 and 4." as you say at the end of the previous section. Maybe you can just shift this sentence to the beginning of this section.**

To address this reviewer's comment and to avoid such confusions, we added Section 4 **Reconstruction of 2D fields** where we mentioned explicitly: *Finally, this algorithm for a single point in time is subsequently applied to all time points of the entire data set shown in Fig. 2, 3 and 4. Thereby two dimensional time-altitude fields of GW parameters can be reconstructed, which is demonstrated in the next section.*

**P12, L1-8: I recommend to put this paragraph prior to the up/downward discussion of literature.**

Changed as suggested.

**P12, L6: Any physical explanation for this finding? Enhanced vertical wavelength due to high wind speed?**

We appreciate the reviewer's analytical reasoning and agree that a more in depth study with emphasis on physical interpretation is needed. However, as we noted already above, we decided to publish such study in a separate paper and involving a lager data set.

Regarding this particular reviewer's point, we can speculate e.g., that in the regions where background wind is very strong the linear theory used in our study fails to describe GW properly. More specifically, our algorithm does not find GWs in this region. This finding has to be studied in more detail before we start to argue for any particular reason.

**P12, L10: No scaling of the amplitudes? To enhance the visibility at lower altitudes compared to higher altitudes, it may be useful scale amplitudes in**

It gave us great pleasure to see that the reviewer appreciates our scaling approach which could also be appropriate in this case, as noted by the reviewer. Nevertheless, we decided to show the reconstructed fluctuation in real physical units (K) to make it easier for experimenters to compare this result with their measurements and to get better filling of the output expected from such analysis.

**Fig. 10. P14, L14-15: But didn't you mention earlier that the sensitivity of your analysis to the chosen background is small?**

**P14, L17: compare comment P6, L19 P14, L21: Wright et al 2017, see major comment 2**

Addressing these two points together, we improved the wording to make it clear in the manuscript that:

1) Among different techniques for background removal we give preference to the 2D-FFT method.

2) Our algorithm to detect GWs is so insensitive to the particular background removal scheme (which is opposite to common knowledge and practice) that it is enough to simply extract a mean value (e.g., to decrease computational load/time)

See also reply to the major comment 2.

**P15, L12: "additional robust algorithm to pick out wave packets automatically". Isn't this what your algorithm does already as implicated by "our algorithm resolves many more GWs than it can be inferred by manually applied hodograph technique"? Please clarify.**

To address this comment and to avoid similar confusions we slightly extended the summary to make it clear that our technique is automatized in spatial domain and not yet in time domain.

*Another specific feature of our analysis technique is the extension to the linear wave theory introduced in Sec. 3, the wave packet envelop term $\exp(-(z-z_0)^2/2\sigma^2)$ that accounts for limited presence of the GW-packet in observations. This, however, only works in spatial domain, i.e. vertically. At the current stage of development our analysis technique is not capable of detecting life-time of gravity waves in observational data set. This capability is currently under development as well as an additional robust algorithm to pick out wave packets in time domain automatically.*

**P15, L14: holographs should be hodographs**

Corrected.

**To Referee 2**

**To test this approach, I suggest the author combine the reconstructed upward and downward perturbations for temperature and wind fields in Figure 10 and 11, and compared with the real perturbations shown in Figure 2,3,4. The total reconstructed perturbations should be quite close to the measured perturbation.**

5    To address this reviewer's comment we made the 2D (time vs altitude) plots of the total reconstructed perturbations. As an example, Fig. 1 demonstrates the reconstruction for meridional wind. Original measurements, reconstructed GWs, and the difference between those are shown in the upper, middle, and lower panel, respectively.

[Figure]

**Figure 1.** Upper panel: Observed fluctuations of meridional wind. Middle panel: reconstructed fluctuations. Lower panel: Difference.

As can be seen from Fig. 1, the reconstructed GW-field (middle panel) resembles the picture formed by the measurements (upper panel), as far as GW-structures are concerned. The lower panel reveals very small-scale noise as it is expected from data

10   treatment and is described in the manuscript. We should admit, however, that critical reader could argue about comparability of and similarity in these figures, as well as about meaningfulness of the noise shown in the lower panel of Fig. 1. To make a justified judgment on such figures one needs to have quite some experience in such (e.g., lidar) data analysis. In other words, we think that non-experienced (in data analysis) reader could rather be confused or missleaded by such figure. Also, including

such plot in the manuscript would need to describe and explain vast of details and thereby, defocus the paper. That is why we decided to not include such comparison in the manuscript.

On the other hand, the next reviewer's suggestion looks brilliant and, in our opinion also serves the same purpose.

**It would also be helpful, if the author could test the results in temporal domain by looking at the wavelet (or lomb-scargle) results (in time) of the real perturbation and the total reconstructed perturbation, to see if the algorithm does not lose the temporal variations of these waves.**

To address this comment and, as mentioned above we believe it supports also the idea in the previous point, we added Fig. 13 with to spectra to the manuscript, as well as a short discussion.

*Another way to check the consistency of our technique is to look at the spectrum of fluctuations before and after analysis. As an example, Fig. 13 shows Fourier spectra of the temperature fluctuations calculated in time domain. The measurements and analysis results are represented by blue and orange lines, respectively. We recall that the analysis is made in spatial domain, that is it only deals with altitude profiles of fluctuations. Close similarity in both spectra which were calculated in time domain, that is across the analyzed profiles, suggests that the reconstructed two dimensional (time vs altitude) GW-field does not significantly deviate from the observed one. The reconstructed field indeed reflects the main GW-content and, therefore, in this respect it may be qualified as lossless algorithm.*

[Figure]

**Figure 13.** Fourier power spectra of measured temperature fluctuations (blue) and of the reconstructed GWs (orange).

**I would also like to know the measurement uncertainties during this lidar campaign, although I am aware that the author treats (weights) every lidar measurement the same (without error?). This is a numerical technique based upon lidar observations, so, I think it is important to know the data quality.**

We completely agree with the reviewer that it would be great to derive valid uncertainties when analyzing experimental data. However, the error propagation issue is not fully addressed in our manuscript since algorithm includes many different data analysis techniques among which the error derivation methods are not well established. So, for instance, is the 2D-FFT analysis or the background removal procedure itself. Frankly speaking, we do not have a clear idea how to estimate uncertainties introduced at some steps of our analysis. Thus, for instance, we can quite precisely derive the uncertainty for every single fit of

the harmonic waves (step 4.4), but we cannot even approximately estimate the uncertainty regarding how precise (or applicable) is the fitted linear model for any particular observation, or how significant is the part for which the fit did not converge (for whatever reason). In other words, even though we are working on this issue, at the moment it does not look convincing to us to discuss the errors related to the reconstructed GW-field in this manuscript.

5 We can comment, however, on the question how measurement errors affect this data analysis. Certain steps in our analysis algorithm directly deal with the measured quantities. These are the places where the uncertainty propagation mathematics can be directly applied. Thus, e.g. as mentioned above we can derive fitting error if measurement errors are known. However, we do not see that these errors can enlighten real uncertainties related to entire analysis.

 Also, to our knowledge, there is no e.g., rigorous mathematical theory to describe error propagation through hodograph 10 analysis. This, we believe, must account also for errors connected with sampling rate and its relation to the eigenfrequencies of the system (GW-filed) under investigation. The similar problem arises also for spectral analyses.

**To Referee 3**

**1) GW Polarization**

The authors present gravity-wave polarization relations, relating zonal and meridional wind fluctuations (Eqn 2), and temperature and zonal wind perturbations (Eqn 3). In equation 3, the authors claim to use the follow Hu et al. (2002) ($Hetal02$) **and Geller and Gong (2010)** ($G\&G10$). **In** $Hetal02$ **the authors have a** ($1/2H$) **term added to the** ($I'm$) **term. In** $G\&G10$ **the authors suggest that their derivation of a polarization relationship relating relative temperature perturbations to pressure perturbations is based on the assumption that the relative temperature fluctuations are identical to the relative potential temperature fluctuations. Do the authors know how these relationships compare to the formulation based on the ideal gas law that relates relative pressure, density, and temperature perturbations directly? If the authors find insignificant differences, particularly for the inertia-gravity waves in this study, then they could explicitly state that.**

Eq. 3 from Hu et al. (2002) is:

$$\widehat{T} = H[im + 1/(2H)]\frac{\widehat{\omega}^2 - f^2}{\widehat{\omega}k_h R} \cdot \widehat{u} \tag{2}$$

where $H = kT_0/mg = RT_0/g$ is scale height. Thus, Eq. 2 can be rewritten:

$$\frac{\widehat{T}}{T_0} = [im + 1/(2H)]\frac{\widehat{\omega}^2 - f^2}{g\widehat{\omega}k_h} \cdot \widehat{u} \tag{3}$$

if therm $1/(2H)$ is neglected, we derive equation used also in Geller and Gong (2010):

$$\widehat{T} = \frac{imT_0}{g}\frac{\widehat{\omega}^2 - f^2}{\widehat{\omega}k_h} \cdot \widehat{u} \tag{4}$$

Indeed, we use an assumption that $\theta'/\overline{\theta} = T'/\overline{T}$. Actually, it is better to use Eq. 7.36 from Holton (2004): $\theta'/\overline{\theta} = \rho'/\rho_0$.

If gravity wave advects air parcels adiabatically, it has a small displacement amplitude ($\Delta z/H \ll 1$) and produces a negligible pressure perturbation within displaced parcels (Fritts and Rastogi, 1985; Eckermann et al., 1998), than we can use equation:

$$\frac{\widehat{T}}{\overline{T}} = \frac{\widehat{\theta}}{\overline{\theta}} = -\frac{\widehat{\rho}}{\overline{\rho}} \tag{5}$$

on the other hand, vertical displacement is related to temperature fluctuation as:

$$\Delta z^2 = \left(\frac{g}{N^2}\frac{\Delta T}{T_0}\right)^2 \tag{6}$$

As can be seen from Fig. 10 of our manuscript, amplitude of the obtained temperature fluctuations is less than $10\,\text{K}$. If we use $T_0 = 253.5\,K$, $N^2 = 3.825 \cdot 10^4\ s^{-2}$, $g = 9.7\ m/s^2$, we obtain vertical displacement equal to 1 km, that is less than H.

**2) Intrinsic and Observed Frequencies**

The authors have complete wind measurements that allows them determine both the observed and intrinsic frequencies of the waves. Can the authors add the observed frequency of the waves to the list of results for the three waves in

**Table 1. In general, can the authors comment on the relationship between the observed and intrinsic frequencies for the waves they have characterized.**

To address this reviewer's comment we extended the Table 1 by adding the observed wave period. The observed period is estimated from the equation:

$$\widehat{\omega}_{intr} = \widehat{\omega}_{obs} - \overline{k} \cdot \overline{u}. \tag{7}$$

It is worth noting here, that another way to derive the observed period from such observations is to apply e.g., Fourier transform to the measured time series. Before that, one has to define the time period during which the wave is present in the observations. This, in turn, can be done (and usually is done so) "manually", by estimating the wave presence by eye. Also, our algorithm in its current status does not allow to estimate the duration of a wave event. We note that it will be difficult (or rather impossible) to find all three waves summarized in Table 1 by this technique.

More specifically, the wave 1 (Table 1), which propagates downward in the same direction as the background wind reveal a period of 38 min. With 15 min temporal resolution such high frequency fluctuations are smeared out in the data.

Wave 2 propagates against wind with phase speed of 22.2 m/s. The wind component along wave propagation is 24 m/s, i.e. is larger than phase speed of this GW. As a result, such wave will reveal 80 h period with upward propagating phase lines. This will appear in the data as horizontal lines not reminiscent of GW.

Wave 3 also propagates upward against the background wind, but its phase speed is higher than the wind speed in the direction of propagation. As a result, the observed period is $19\ h$. Similar to the wave 2, this will be hardly resembling the GW and, therefore, rather not detectable.

Our analysis technique based on hodograph, in turn, allows to detect such long period waves by utilizing the Eq. 7. Left panel of Fig. 10 demonstrates that the most of the detected waves reveal large periods. Periods of downward propagating GW is more difficult to estimate even from such reconstructed time series because they are essentially not continuous.

We extended discussion of these issues in the revised version of manuscript to properly address this reviewer's comment.

**3) Identifying Waves**

**The study reports 4507 quasi-monochromatic waves. However, if I understand it right the study has found 4507 snapshots of some number of waves based on hodograph analysis of 240 profiles. The authors discuss that the individual waves persist in their presentation of temperature fluctuations and intrinsic periods in Figures 10 and 11. Can the author quantify the life-time of the waves in the data set? There have been discussions in the literature about how many gravity waves are present and the intermittency of gravity waves. This study has the opportunity to address the life-time of gravity waves, particularly relating it to the spatial and temporal scales of the waves, that would address a variety of questions about wave dynamics and evolution.**

The reviewer is absolutely correct that we found 4507 snapshots of some number of waves based on hodograph analysis of 240 profiles. We also agree that the scientific questions pointed out by the reviewer in this comment are of a high interest and importance.

However, as we already mentioned in our response to the previous comment, at the current stage of development our analysis technique is not capable of detecting life-time of gravity waves in observational data set. We are working on this and anticipate some progress in the nearest future. At the moment we can only make some statistics and analyze the relations between different GW-parameters and e.g., amplitude of wave as a function of background wind.

Fig. 10 and 11 from the manuscript represent our first attempt to analyze temporal development of the detected wave packets. Thus, for instance, on Fig. 11, one can recognize regions of near the same color which resulted from many adjacent profiles that reveal very close intrinsic frequencies. This can be picked up by the naked eyer. We can assume that such regions depict propagation of the same wave packets. However, a more in depth analysis must be performed (that probably should utilize some additional criteria) to developed a more robust algorithm to pick out wave packets automatically.

At the moment we decided to characterize GW based on so far well established combinations of wave parameters. Thus we can select waves, for example, that propagate in a given direction (up or down) and horizontal wavelength in some fixed range of values and reconstruct pictures like Fig. 10 and 11 for speculative analysis.

To address this reviewer's comment we added a short discussion of this issue in our manuscript.

**To Referee 4**

**You produce most of your diagrams for "number of waves". However, from the dynamics point of view GW pseudo-momentum flux is most relevant. It would be very helpful if you add a second row to Figure 12 where you plot the total absolute momentum flux of the waves in a wavelength bin. (You could normalize that in a way that the total GWMF of all waves (up + down) is normalized to 1. and keep that same normalization also for up and down separately). Same for F14 and F15.**

We appreciate the reviewer's suggestion and interest in seeing more geophysical results. We decided, however, to limit ourself in this paper to the technical questions of derivation of GW parameters. We also aim at doing a more in depth geophysical study based on a lager observational dataset, which in particular also covers different seasons. We will definitely address these questions in that paper.

In what follows, however we try to show to reviewer what can be inferred from this limited set of measurements.

First of all, to our understanding, the requested by the reviewer "total absolute momentum flux of the waves" is exactly the quantity which we show in our manuscript in Figs. 17 and 18 (18 and 19 in the revised version of the manuscript).

The absolute momentum flux was estimated, for example, by Ern et al. (2004); Ern et al. (2016), and can be written in form:

$$F_{Ph} = \sqrt{F_{Px}^2 + F_{Py}^2} \tag{1}$$

We show observed momentum flux in given direction ($F_{P\parallel}$). That is, we understand that the Ern et al.'s absolute momentum flux is the same as our reported momentum flux:

$$F_{Ph} = \sqrt{F_{P\parallel}^2 cos(\xi)^2 + F_{P\parallel}^2 sin(\xi)^2} = F_{P\parallel}\sqrt{cos(\xi)^2 + sin(\xi)^2} = F_{P\parallel} \tag{2}$$

To address this reviewer's question, we derived some additional dependencies and show the results here.

First, in Fig. 1 we show the total absolute momentum flux of the waves as inferred from our analysis. Small dots show results derived for every successful hodograph analysis. Colored lines show an average momentum flux for up- and downwards propagating GWs separately.

Next, a momentum flux in east-west direction as a function of vertical wavelength is shown in Fig. 2. Momentum flux is positive if waves propagate towards east.

It is important to mention, that momentum flux depends not only on vertical wavelength, but also on horizontal wavelength as demonstrated in Fig. 3.

Finally, the total absolute momentum flux of the waves in a 100 km wavelength bin is shown in Fig. 4.

Nonetheless, as mentioned above, we believe that including such figures in the manuscript, will defocus the paper from methodological to scientific emphasis, which contradicts our current goal.

**The vertical wavelengths you observe are rather small. Starting from the very first work on saturated spectra (Smith, Fritts VanZandt, 1987) we have indication that the wavelength of the maximum in the distribution shifts to longer wavelengths at higher altitudes. Follow-up work by e.g. Gardener et.al. and the general concept of the Warner &**

[Figure]

**Figure 1.** Absolute momentum flux. Red (blue) line and dots marks upward (downward) propagating GW.

[Figure]

**Figure 2.** Momentum flux in East-West direction

**McIntyre scheme infer a power law for this. You can put in several observations by e.g. radio sondes, rockets ... to calibrate this. Then you would expect something like 2 km in the lower stratosphere, 10-15 km in the mesopause region and accordingly $\sim 5$ km around the stratopause. The satellite data certainly have a long-bias, but they confirm the increase of typical wavelengths with altitude. Compared to this you have 2 km which one would expect for the low stratosphere in a data set which goes up to the mesopause. One reason may be that you give your histograms for number of waves only. Still it would be good to see some vertical profile of avereage vertical wavelengths, normal average as well as GWMF weighted, up + down separately, so for profiles in total.**

[Figure]

**Figure 3.** Momentum flux

[Figure]

**Figure 4.** Momentum flux averaged in 100 km bins

We appreciate this reviewer's constructive comment and totally agree to properly address this question in our next paper (see also our general comment and reply to reviewer's comment 1). Here, again we show what can be inferred so far from this first, limited (in sense of data coverage) observational set.

To address this reviewer's question we split the data shown in the leftmost histogram in Fig. 12 of our manuscript in several altitude bins. The result is shown here in Fig. 5. One can see, that distribution gets broader with increasing altitude which is consistent with the increase of the wavelength pointed out by the reviewer. Several following reasons, however, prohibit from drawing strong conclusions. First, upper and lower boundaries of our observational domain cannot include longwavelength-waves due to limitation of the analysis techniques (we require that the wave packet is almost completely present in the observations, vertical wavelengths longer than 15 km are partly attributed to the background and, therefore, are not considered). Second, this statistics ultimately includes all kind of GWs including secondary, tertiary, whatsoever appears in the atmosphere, whereas reviewer's argument might only refer to the waves propagating from the ground (or troposphere).

**Phase speed is approx proportional to vertical wavelength. The observational filter for airglow is totally different (lz>10km), so no wonder that phase speeds are much higher. There is a wealth of literature on phase speeds from different sources (convection, spontaneous imbalance, ...). Maybe it is more worthwhile to compare to that. The phase speed diagram kind of seems to exclude convection as dominant source here. Still there is the general issue about the short vertical wavelengths.**

Again, we appreciate the the reviewer's valuable suggestion which we plan to address in our next work with more detailed geophysical analysis. The sources of GWs with different characteristics is for sure of a great scientific interest. For this particular purpose we additionally involve analysis of different simulation data which, we believe must shad some more light on this question, than simply speculate based on limited observational base.

Minor comments and technical suggestions:

**P2L20 Did the Shigaraki radar not provide some winds? If so: provide high-resolution wind**

All MST radars are not capable of measuring in the altitude range $\sim$30 to $\sim$60 km because of the absence of suitable scatters. As we can judge from the description in the WebSite

$(http : //www.rish.kyoto - u.ac.jp/organization_e/collaborative_research/mur/)$,

Shigaraki MU Observatory also provides lidar observations in the altitude range 30 to 60 km.

**P7L17 So you don't do that? Why not?**

Zink and Vincent (2001) and Murphy et al. (2014) used sum of scalograms of both wind components. We used product of scalograms of all three components, i.e., u, v, and T.

A product of spectra works similar to Cross Wavelet Spectrum (Torrence and Compo, 1998) with difference that it allows to compare three spectra simultaneously. It will only reveal an enhanced power where all spectra under analysis show high power.

As an example, if some wave with large enough amplitude is only presented in one component, but absolutely absent in all other components, the sum of scalograms will show signature of this wave. Our purpose is to detect regions where all three components (u, v, T) reveal wave oscillation. The product, in turn, will show low power. E.g., for u=0, v=0, T=1 product gives 0, whereas sum is equal to 1.

**F13 I like that figure, but it would be great if you could add two more panels: Vertical wavelengths and GWMF.**

Here we have to refer to our major comment. In particular, we do not see any dependence of vertical wavelength on altitude in this dataset. Also, to our understanding, the requested GWMF is shown in Fig. 17 (Fig. 18 in the revised version of manuscript).

**P14L1 And this is really puzzling! You have most of the waves and the momentum flux propagating against the wind and the wind velocity increases at higher altitudes, so no critical level filtering. Vertical wavelengths then should increase which leads to lower amplitudes at same GWMF, so no saturation expected either. Reflection? It would be good to know**

[Figure]

**Figure 5.** Histograms of observed vertical wavelengths. Different altitude ranges are shown in different plots.

**at least which parameter changes most (wavelengths, amplitudes ...) as to produce this result. Or do you have an edge effect in your retrieval or your method?**

We recall, that the chosen data set does not represent a typical picture at observational site. Moreover, we found, that during the period of observations presented in this work strong polar vortex was observed right above the ALOMAR observatory. Also, two upper panels in Fig. 3 of the manuscript reveal zonal wind higher than 100 m/s between 40 and 50 km. This suggests, that most likely the majority of detected waves was generated in stratosphere.

**All other minor comments and suggestions were implemented as suggested.**

[revised manuscript text omitted]

---

## Referee Report (RR1)

**Review of „Advanced hodograph-based analysis technique to derive gravity waves parameters from Lidar observations" by Strelnikova et al.**

The structure and readability of the paper was improved in the latest Version of the Manuscript. The content and the way it is presented are suitable for publication in AMT. However, the scaling part of the analysis still needs revision in my opinion before begin published. See major comment I.

Major:

I)      Scaling: In your response you stated that your parameter in the scaling function is determined for every individual profile. However, it's stated on page five that it is 2.15. On P8, you say that you get rid of the exponential growth using scaling. Eq. 6 comes without exp(z/(2H) and you refer to your scaling which is exp(z/(2.15*H)). This does not seem to work out (or I am getting it wrong). If you don't have strong reasons for doing the scaling the way you currently do it, you should simply use exp(z/(2H)) and refer to Wright et al. as you do later in the text. This topic needs to be revised in the Discussion section as well.

Minor:

I) Before being published, the author's should carefully revise language, sentence structure and typos at several locations in the manuscript (e.g., "Although, any and also vertically propagating waves might appear in the nature in the form of wave packets rather than continuous wave of quasi-infinite length."; "Tides periods that are integer fractions of a solar day.",…)

P3, L1: order of the outline (section 3 before 4,5…)

P4,L22: Define/describe F?

P5,L12: Name here the other characteristics that are determined.

P5,L19-L24: I would not call it bias, rather differences between the different approaches for determination of the background.

P6,L4: change order of the Figures (start with current Fig. 2).

P7, L2: Should be (a) 2D-FFT, (b) running mean, …..

P7, L5: You stated earlier that the background removal is critical and here you say your methods could work without this step? Then why not skip it in the first place? If you are no doing it, then better remove this sentence.

P9, L16: It's only a circle for frequencies *equal to* the Coriolis frequency. For low frequency GWs the hodograph forms an ellipse.

P10, L21: Name the characteristics here.

P11, Section title, L6: Reconstruction, better use derivation/determination instead of reconstruction

P13, L32: why likely? They are assigned to the background by the method.

---

## Author Response (AR2)

**Response to reviewers' reports on the revised version of paper amt-2019-79**

Irina Strelnikova[1], Gerd Baumgarten[1], and Franz-Josef Lübken[1]

[1]Leibniz-Institute of Atmospheric Physics at the Rostock University, Kühlungsborn, Germany

*Correspondence to:* Irina Strelnikova (strelnikova@iap-kborn.de)

We appreciate the reviewers' constructive comments, their careful reading with attention to details, and their positive judgment on our paper. We have taken the reviewers' suggestions into account when preparing the revised version of our manuscript.

In the following we address the comments of all reviewers point by point.

**To Reviewer 1**

5 The major reviewer's suggestions were:

**2. page 5 line 29, the author chooses 15 km as the upper limit. Can you justify it? The diurnal tide has the vertical wavelength > 20 km, why not choose 20 km as the upper limit?**

To address this reviewer's comment we rephrased the sentences to make it clear that the 15 km limit only applied to waves with periods >12 h.

10 *Thus, we define the background as wind or temperature fluctuations with periods longer than 12 hours and vertical wavelengths larger than 15 km. Fluctuations with periods shorter than 12 hours, which have any vertical wavelength (also grater than 15 km), are attributed to GWs and are the subject of further analysis.*

**3. On page 7, in line 10. "a constant background" is misleading, I think. The background is not constant over time, so I think the author is referring to an instantaneous constant vertical background profile. Please clarify.**

15 We replaced the sentence *"a constant background"* by a more precise one:

*isothermal atmosphere without background winds*

**4. On page 9, line 25-27. Please clarify why removing waves with vertical wavelength less than half of the wavelength and twice the wavelength can minimize the uncertainty due to the presence of the other waves.**

To address this reviewer's comment we added the following explanation in the manuscript:

20 *Such a filtering (and especially its high-frequency part) works as a denoising for the profiles and improves the robustness of the subsequent fitting. The choice of the filter width ($\lambda_z/2$, $2\lambda_z$) is rather arbitrary and e.g., can be inferred from e-folding time of the wavelet function around the detected peak.*

All other corrections were implemented as proposed.

**1   To reviewer 2**

**Scaling: In your response you stated that your parameter in the scaling function is determined for every individual profile. However, it's stated on page five that it is 2.15. On P8, you say that you get rid of the exponential growth using scaling. Eq. 6 comes without exp(z/(2H) and you refer to your scaling which is exp(z/(2.15*H)). This does not seem to work out (or I am getting it wrong). If you don't have strong reasons for doing the scaling the way you currently do it, you should simply use exp(z/(2H)) and refer to Wright et al. as you do later in the text. This topic needs to be revised in the Discussion section as well.**

To address this reviewer's comment and, thereby to avoid misunderstanding, we changed the text. We believe that it is now clear from the text that:

1. introduced scaling is essential for detection of GWs at lower altitudes

2. our scaling accounts for real growth of GW-amplitudes via factor $\varsigma$

3. real $\varsigma \geqslant 2$, which is supported by references to theory and observations

4. after the scaling is applied to profiles, the wave-function (Eq. 2) must be changed accordingly: i.e. exponential term must be dropped (Eq. 6)

5. this scaling has to be removed again to analyze energy content of the timeseries

*Section **Fitting of linear wave theory**.*

*After applying the scaling $1/\exp(z/(\varsigma H))$ to fluctuation profiles as described in Sec. 4.2, we get rid of exponential growth in those profiles. Therefore, we have to exclude this scaling factor from the wave equation. The wave-function that we fit to the profiles $u'(z), v'(z)$, and $T'(z)$ reads:*

$$\vartheta' = |\widehat{\vartheta}| \cdot \exp(-(z-z_0)^2/2\sigma^2) \cdot \cos(m(z-z_0) + \varphi_\vartheta) \tag{6}$$

*where $\vartheta$ refers to $u, v$ and $T$.*

*Section **Results and discussion***

*First, a short discussion about the exponential scaling factor $1/\exp(z/(\varsigma H))$ applied to the fluctuation profiles as described in Sec. 4.2 has to be made. This factor should compensate for the exponential growth of GW-amplitude due to the exponential decrease of atmospheric density. It is commonly accepted to use $\varsigma = 2$. However, since the exact value of $\varsigma$ depends on the particular state of the atmosphere during the observations, it has to be directly estimated from the measurements. Thus, e.g., Fritts and VanZandt (1993) theoretically derived $\varsigma = 2.3$ consistent with number of observations revised in e.g., Fritts and Alexander (2003). Lu et al. (2015) incorporate this factor into the "observed" scale hight which imply $\varsigma = 2.5$ to $2.8$ for different observations over the McMurdo (77.8 °S, 166.7 °E). We derived $\varsigma = 2.15$ as a mean value over the entire time series, that is as an average of $\varsigma$ of all the individual profiles. We assume that $\varsigma$ does not change significantly during the observational*

*time period of approximately three days. However, if observations will last longer, this assumption will not hold. In such a case the scaling function has to be optimized to reveal some time dependence (not addressed in this work).*

**P5,L19-L24: I would not call it bias, rather differences between the different approaches for determination of the background.**

We believe that as far as the background is defined in a way we do it in our manuscript, it should be quantitatively described by corresponding temperature and wind fields. This quantification must be unique, i.e. must have its true value. This true background, of course, can only be measured with certain precision (i.e. error). We believe, that the different approaches for determination of the background discussed in our work unavoidably introduce systematic errors, which we gently called bias.

**P6,L4: change order of the Figures (start with current Fig. 2).**

Unfortunately, we did not understand this comment, since the order of Figures in the text starts with Fig. 2

**P7, L2: Should be (a) 2D-FFT, (b) running mean,**

To address this reviewer's comment we added some clarifications in the text to make it clear, that the new list of the background derivation methods was an addition and only aimed to be a robustness check.

*To derive the background (for both wind and temperature data) we **additionally** made use of (a) running mean with different smoothing window lengths, (b) different splines, and (c) constant values in time.*

See also the next point.

**P7, L5: You stated earlier that the background removal is critical and here you say your methods could work without this step? Then why not skip it in the first place? If you are no doing it, then better remove this sentence.**

To address this referee's comment we extended the explanation for the necessity and advantage of the background determination by the 2D-FFT technique.

*Even though we are confident in the robustness of our GW-analysis technique to the various background derivation methods, we need a well-defined and well-behaved (i.e. continuous and smooth) background (1) to derive the basic parameters of atmosphere like buoyancy frequency and wind shear and (2) to find out how the background wind and temperature fields affect (or at least correlate with) the GW-field.*

**P9, L16: It's only a circle for frequencies equal to the Coriolis frequency. For low frequency GWs the hodograph forms an ellipse.**

To address this referee's comment we changed the phrasing to be more precise as suggested by the reviewer.

*For low frequency GWs, i.e. those with periods close to the Coriolis period $(2\pi/f)$ the fluctuations' **hodograph closely resembles circle**.*

**P11, Section title, L6: Reconstruction, better use derivation/determination instead of reconstruction**

We are convinced that the term ´´Reconstruction" is more appropriate in this case.

**P13, L32: why likely? They are assigned to the background by the method.**

To address this reviewer's comment we made it clear in section 4.1 **Separation of GW and background**, that the waves with $\lambda_z > 15$ km are defined as a background only if their period >12 h. If their period is less than 12 h, they are not excluded from the analysis and remain in the fluctuation profiles after the background removal (if 2D-FFT method is used).

[revised manuscript text omitted]